# An in vitro model maintaining taxon-specific functional activities of the gut microbiome

Leyuan Li[1,4], Elias Abou-Samra[1,4], Zhibin Ning[1], Xu Zhang [1], Janice Mayne[1], Janet Wang[2], Kai Cheng[1], Krystal Walker[1], Alain Stintzi[1] & Daniel Figeys[1,3]

In vitro gut microbiome models could provide timely and cost-efficient solutions to study microbiome responses to drugs. For this purpose, in vitro models that maintain the functional and compositional profiles of in vivo gut microbiomes would be extremely valuable. Here, we present a 96-deep well plate-based culturing model (MiPro) that maintains the functional and compositional profiles of individual gut microbiomes, as assessed by metaproteomics, while allowing a four-fold increase in viable bacteria counts. Comparison of taxon-specific functions between pre- and post-culture microbiomes shows a Pearson's correlation coefficient $r$ of $0.83 \pm 0.03$. In addition, we show a high degree of correlation between gut microbiome responses to metformin in the MiPro model and those in mice fed a high-fat diet. We propose MiPro as an in vitro gut microbiome model for scalable investigation of drug-microbiome interactions such as during high-throughput drug screening.

[1] Department of Biochemistry, Microbiology and Immunology, Ottawa Institute of Systems Biology, Faculty of Medicine, University of Ottawa, Ottawa, Canada. [2] Department of Statistical Sciences, Faculty of Arts and Science, University of Toronto, Toronto, Canada. [3] Canadian Institute for Advanced Research, Toronto, Canada. [4] These authors contributed equally: Leyuan Li, Elias Abou-Samra Correspondence and requests for materials should be addressed to D.F. (email: dfigeys@uottawa.ca)

The gut microbiota is being increasingly recognized as a key factor in human health and disease[1], and as such, it is an important target for drug therapy[2]. Evidence is mounting that microbial metabolism of drugs can directly affect drug efficacy and toxicity[3,4], and that drugs can, in turn, alter the composition and function of the gut microbiota[5–9], potentially affecting the health of the host. Therefore, the gut microbial ecosystem, specific microbes, and microbial pathways are novel targets in drug discovery.

In vitro culture models could be a timely and cost-efficient way to discover microbiome responses to drugs. However, current culture models do not maintain the functional and compositional profiles of the gut microbiome. To simulate an in vivo microbial ecosystem, it is key to conserve both the composition and functional activities of an individual's microbiome. In current in vitro culturing methods, profound shifts in taxa proportions have been frequently described during both batch culturing[10,11] and continuous flow culturing models[12,13] when compared to the inoculum. Importantly, shifts in microbiome composition can alter their functional properties and ecological processes[14], which could affect microbial responses to drug stimuli. To the best of our knowledge, there have been no reported models that maintain microbiome properties similar to the inoculum. In particular, the preservation of functional activities has not been described elsewhere. Moreover, a high percentage of marketed drugs, and compounds in development, may have off-target effects on the gut microbiome[15,16]. Maier et al. observed that non-antibiotic drugs had extensive antibiotic-like impacts on cultured human gut bacterial strains[17]. However, the response of isolated strains may differ from that of a functionally preserved gut microbiome due to the complex functionality of a microbial community. Therefore, systematic studies of drug effects using functionally maintained individual gut microbiomes, as the in vitro model, is a pressing need. For long-term observations of xenobiotic effects, continuous flow systems[10,12,18] and microfluidic models[19,20] work well. However, these models can not be readily adapted for high-throughput approaches, partially due to their sizes and the time required for setting up and stabilizing these bioreactors.

Gut microbes can respond to altered environmental factors within a single day[21–23]. To gain a deep insight into a microbiome's responses to drugs, a technique that can precisely quantify microbial functional activities is required. Metaproteomics is a meta-omic tool that directly quantifies the microbial functional responses at the end-product level of expressed proteins[24]. The development and application of mass spectrometry (MS)-based metaproteomic technologies in gut microbiome research has thrived in recent years. The identification coverage and sensitivity of MS-based metaproteomics have increased dramatically, enabling in-depth analysis of microbiome functional activities[25,26]. An advantage of metaproteomics when compared to sequencing-based techniques is that it can combine functional and phylogenic information. Metaproteomics is also more accurate for biomass estimates on a species level[27], and it enables the study of taxon-specific functions through annotation of unique peptides[28]. Comparison of taxon-specific functional profiles of the microbiome is important for determining cultured stability and drug responses. This is because overall functions can remain relatively stable due to the redundancy of functional genes among species in a gut microbiome[29]. As well, genes predicted from metagenomic analyses are not necessarily expressed. Therefore, metaproteomics is a suitable tool for insights into in vitro drug responses of gut microbiome.

Here, we report the development and validation of a scalable in vitro model for the maintenance of gut microbiome profiles (MiPro). Briefly, the MiPro model adopted an optimized culture medium and a 96-deep well plate-based format for microbiome

culture (Fig. 1a). The culture medium and culturing conditions were improved from our previous medium composition study[22]. The model was first evaluated for its ability to maintain gut microbiome profiles in vitro, followed by testing and evaluation of the model's in vitro correlation with in vivo drug response. This culture model enabled, in combination with metaproteomic analysis, the assessment of drug effects on the microbiome at compositional and functional levels. This model maintained high microbiome compositional stability and greater than 0.83 taxon-function similarity (Pearson's $r$) over 24 h. In addition, we demonstrated that our MiPro model recapitulated the in vivo effects of metformin observed on individual mouse gut microbiomes.

## Results

**Establishment of the MiPro model**. The first objective of this work was to develop a scalable in vitro model for the maintenance of gut microbiome profiles. We have recently evaluated the composition of the culture medium for optimal culturing of ex vivo microbiomes[22]. Here, we improved upon the composition of our previously reported medium by assessing the effect of bile salts formulae on the gut microbiome. Two formulae were compared: (1) a mixture of primary bile salts, i.e. 1:1 (w/w) sodium salt forms of cholic acid (CA) and chenodeoxycholic acid (CDCA), and (2) a commercial 1:1 (w/w) mixture of sodium salt forms of CA and deoxycholic acid (DCA). The commercial bile salts mixture has been adopted by a majority of studies employing gut microbiome culture media[11,12,30–36]. We compared the maintenance of taxon-specific functional profiles for gut microbiota cultured in the presence of either formulae (Fig. 1b). Briefly, gut microbiota cultured for 24 h were subjected to metaproteomic analysis, and a taxon-function-coupled analysis was carried out using the iMetaLab platform (http://shiny.imetalab.ca/, Supplementary Fig. 1). We observed significant increase of Pearson's correlation coefficient $r$ of taxon-specific functional profiles between the inocula (0 h baseline sample) and the cultured microbiome in the presence of the primary bile salts mixture (CDCA + CA). Moreover, comparison of microbiome composition (Supplementary Fig. 2a, b) and the abundances of clusters of orthologous group (COG) categories (Supplementary Fig. 2C) also suggested that the presence of CDCA + CA could maintain the microbiome better than DCA + CA.

Additionally, we assessed whether the gut microbiome culture was affected by the culturing conditions, namely (1) tube-based and (2) 96-deep well based culturing, while keeping all other conditions, including medium, inoculum, temperature, and container material (polypropylene) constant. Culture tubes are the most frequently used containers in batch culture experiments[11,22,36]. Notably, the 96-deep well was covered with a silicone-gel cover, which was perforated at the top of each well. This cover could impair gas-exchange with the outer environment in the chamber, so as to preserve the partial pressure of gases and volatile metabolites in each well, which could subsequently preserve certain levels of dissolved gas molecules in the culture medium. In contrast to the use of 96-deep well plates, employing culture tubes resulted in a remarkable change in the taxon-specific functional activity (Fig. 1b) as well as other taxonomic and functional profiles (Supplementary Fig. 2). From the above results, we established our MiPro model (Fig. 1a): a 96-deep well plate-based culturing in combination with the optimized medium which contains 1:1 (w/w) of CA and CDCA (hereafter called MiPro medium).

**MiPro increased viable bacteria and quintupled the count.** We next evaluated the ability of the MiPro model to sustain a viable

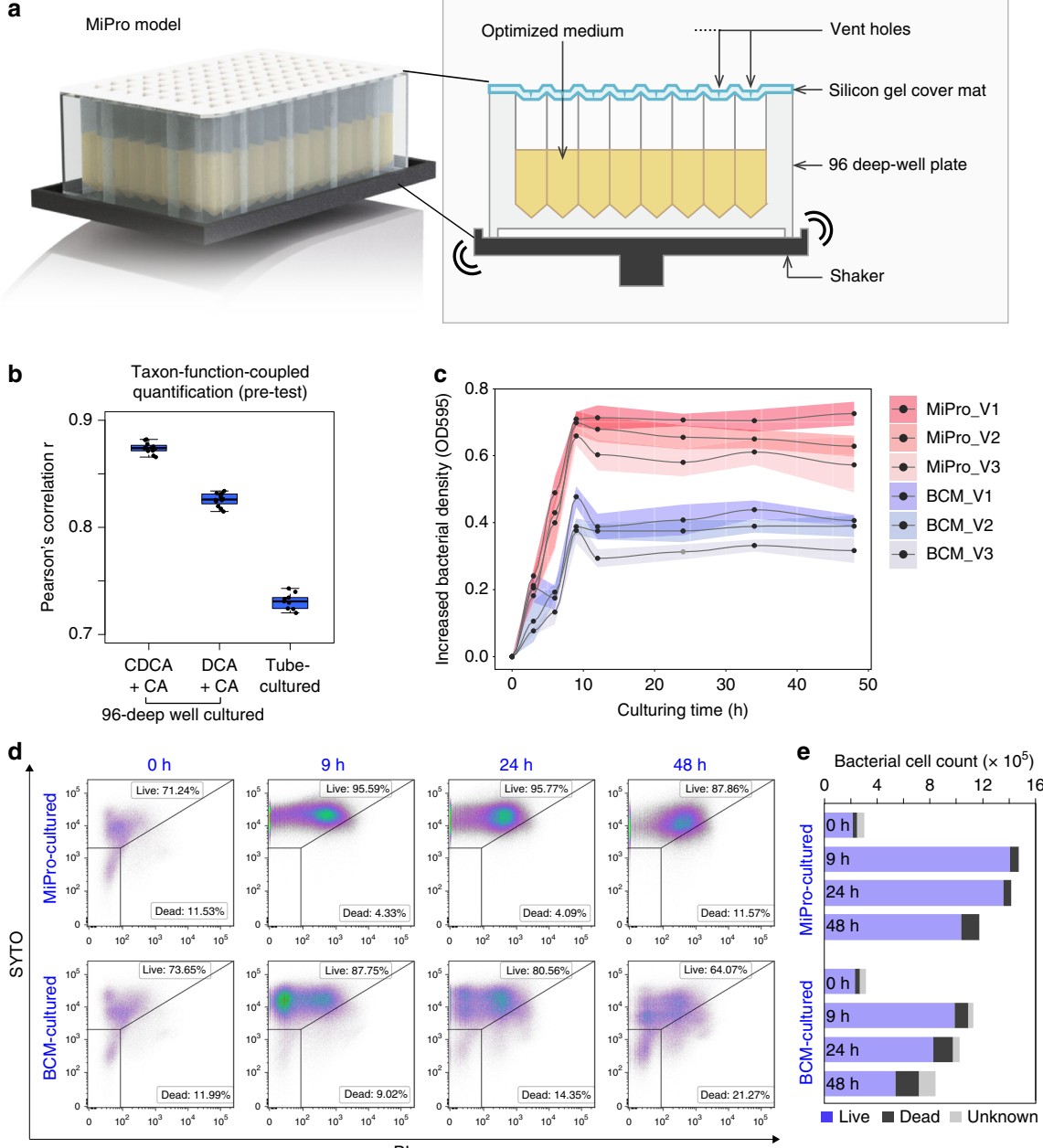

**Fig. 1** Establishment and general performance of the MiPro model. **a** Main components of the MiPro model: microbiome samples are cultured in an optimized culture medium in a 96-deep well plate. The plate is covered with a silicone-gel cover perforated at the top of each well. The plate is shaken at 500 rpm on a digital shaker. **b** Pearson's correlation coefficient $r$ of taxon-specific functional profiles between the inocula (0 h baseline sample) and 96-deep well-cultured microbiome with the presence of primary bile salts (CDCA + CA) or commercialized bile salts mixture (DCA + CA), as well as tube-cultured microbiome with the presence of primary bile salts. Different letters indicate significant differences at the $p = 0.05$ level, Tukey-b test; box spans interquartile range (25th to 75th percentile), and line within box denotes median. Whiskers represent min to max values. $n = 3$ biologically independent microbiomes. **c** Increase in bacterial biomass over time in each individual microbiome (determination using absorbance at 595 nm). Colored ribbons indicate the range of standard deviations around the means ($n = 4$ technical replicates for each treatment). **d** Temporal microbiome viability changes as shown with flow cytometry. The gating strategy is shown in Supplementary Fig. 3. (**e**) Bacterial cell count in each flow cytometry recording. Data were recorded for 2 min on the low flow rate setting. Underlying data are provided in the Source Data file

microbiome, which is vital for an effective in vitro microbiome response study. Temporal changes in parameters including microbial biomass, cell count, viability and diversity were compared with the 0 h baseline sample. The commonly used basal culture medium (BCM) was included for comparison. Since different microbial members can differ by several orders of magnitude in biomass[27], it is necessary to examine both biomass and bacterial counts. We used optical density at 595 nm ($OD_{595}$) to determine bacterial biomass density, and used flow cytometry to determine bacterial cell counts. At 24 h the $OD_{595}$ was $0.8 \pm 0.1$ fold higher (i.e., 80% higher) in the optimized medium as compared to that cultured in BCM (Fig. 1c). Furthermore, using flow cytometry in combination with viability staining, we found that our MiPro medium achieved high ratio of viable bacterial cells throughout the culture period (95.8% at 24 h compared with 71.2% at 0 h, Fig. 1d). Comparison of the flow cytometry readouts

between the inoculum and the cultured microbiome showed a 4.4-fold increase of viable bacterial count in MiPro medium after 24 h of culturing, whereas a 3.0-fold increase was detected in the BCM medium at 9 h post-culturing and followed by a decline.

**MiPro model maintains taxon-specific functional profiles.** In order to evaluate the effectiveness of MiPro to simulate the in vivo features of the gut microbiome inoculum, we used a metaproteomic approach[27,37] to characterize the taxonomic and functional stability of three individuals' gut microbiomes over 9, 24, 34 and 48 h of growth in either MiPro or BCM media. A minimum of three cultured, technical replicates were analyzed at each time point by LC-MS/MS. Ninety high-quality MS raw files were obtained with a total of 2,066,069 MS/MS spectra. With an average MS/MS identification rate of 40.8% ± 4.5% (mean ± SD), a total of 58,848 peptides and 16,326 protein groups were identified with a false discovery rate (FDR) threshold of 1% (Supplementary Fig. 4a). A high concordance (Pearson's correlation coefficient $r = 0.97 \pm 0.02$) was observed between the technical replicates of each group, indicating robust experimental reproducibility (Supplementary Fig. 4b). Using an LCA approach on the MetaLab[37], a total of 21,839 peptides were assigned with a taxonomic lineage, resulting in 788 assigned species. The quantitative information (summed peptide intensities) was used to assess the species-level biomass contributions[27,37]. 121 species that were quantified with ≥3 peptides were included in the comparison of species biomass contributions.

We applied a Bray-Curtis dissimilarity-based approach[38] for evaluating the variation of species biomass contributions between groups (Fig. 2a). All MiPro-cultured microbiomes clustered closely with their corresponding inocula (0h baseline samples). In contrast, the BCM-cultured microbiomes were well separated from their inocula and exhibited greater dispersion over time than the MiPro-cultured microbiomes. Analysis of similarity (ANOSIM) based on the Bray-Curtis distance showed that at all time points, Mipro-cultured samples had an ANOSIM's R of 0.122, $p = 0.025$; while the BCM-cultured microbiome had an R of 0.2892, $p = 0.001$. Moreover, species-level biomass distribution bar-charts showed that the microbiome composition of three different individuals were sustained throughout the culturing period in the presence of MiPro medium (Fig. 2b and Supplementary Figs. 5 and 6). In particular, one-way ANOVA analysis on the latter dataset showed that the average relative abundance of *Faecalibacterium prausnitzii*, one of the most abundant gut microbial species quantified with metaproteomics in this dataset, significantly decreased in the BCM medium in comparison to both MiPro culture and baseline samples (Supplementary Fig. 7). In addition, both MiPro and BCM media achieved well-maintained alpha-diversity (Shannon-Wiener index) overtime in the cultured microbiomes from all three volunteers (Supplementary Fig. 8).

To assess the stability of functional activities, the identified proteins were annotated with COG categories and the abundance of each COG category was calculated by summing the LFQ intensities of all the proteins belonging to the same COG category[39]. Principal component analysis (PCA) was used to assess the relatedness of the samples based on the functional makeup of the metaproteomes. The first two components, PC1 and PC2, explained 71.7% and 21.4% of the total variance, respectively (Fig. 2c). The largest functional variability was found with the BCM medium, indicating that using the MiPro medium results in a better maintenance of the inoculum's microbial functional profile (Fig. 2c and Supplementary Fig. 9). In order to assess the maintenance of taxon-specific functional traits, we carried out a taxon-function-coupled analysis using the iMetaLab

platform (http://shiny.imetalab.ca/)[40]. In total we identified 1,066 unique COGs of proteins corresponding to 419 taxa, the overall taxon-function distribution across the dataset is shown in Supplementary Fig. 1a. By generating the taxon-function distribution for individual samples, a three-dimensional dataset for between-sample comparisons was created (sample-taxon-function, Supplementary Fig. 1b). Pearson's correlation coefficient $r$ of the taxon-specific functional profiles was calculated between the inoculum and the cultured microbiome for each time point (Fig. 2d). Compared with BCM, the MiPro culture achieved significantly higher taxon-specific functional correlations with baseline samples ($p < 0.05$; Fig. 2d and Supplementary Fig. 10). Pre-post culture correlations of taxon-function-coupled profiles reached an average of $r = 0.83 \pm 0.03$ with our model (an average of all time points). To gain a more intuitive understanding of this correlation, we performed a taxon-function-coupled enrichment analysis using iMetaLab. Figure 2e shows the top 30 correlations (determined by the numbers of taxon-function matches) between the taxa and functions of the inoculum microbiome (0 h baseline) and the 24 h cultured microbiome for one individual. In addition, Supplementary Fig. 11 visualizes a comparison of the top 300 correlations. Similar taxa-function profiles were observed between the inoculum and 24 h cultured microbiome indicating functional robustness in the MiPro culture.

We then investigated the maximal culture time that MiPro could achieve. Noting that there would be consumption of medium nutrients and accumulation of microbial metabolites over time that may affect microbiome functions, we replaced one quarter of the cultured suspension (i.e., 250 µl) with same volume of fresh medium every 12 h. Samples were analyzed by metaproteomics at 0 h baseline and at 1-, 2-, 3- and 5-days of growth. We quantified 32,030 peptides and 9307 protein groups from the fifteen samples with an average MS/MS identification rate of 51.2% ± 1.6% (mean ± SD). 29,882 peptides were assigned a taxonomic lineage, yielding 149 species that were quantified with ≥3 peptides. Taxon-function-coupled analysis showed that the Pearson's correlation coefficient $r$ of the taxon-specific functional profiles between the baseline and cultured microbiome were maintained well above 0.8 for 5 days of growth (Fig. 2f). The species-level biomass distribution also suggested that the microbiome was sustained throughout the 5-days of culture (Fig. 2g).

**In vitro–in vivo correlation of drug response.** We then evaluated the in vitro–in vivo correlation (IVIVC) of microbiome drug response in the MiPro model using C57/BL6 mice fed a high-fat diet (HFD). Metformin is a widely prescribed drug for treating type 2 diabetes and it has been reported that in human 30% of the oral dose can be recovered in feces[41]. Several studies have shown that metformin alters gut microbiota composition and functions[5,6,42]. We, therefore, employed metformin to validate our MiPro model by comparing the impact of metformin exposure on gut microbial communities in mice and in our in vitro model. Briefly, the MiPro model was inoculated with the stool microbiome from each mouse and cultured for 24 h in the presence or absence of metformin. Mice were treated daily for 28 days with 300 mg/kg of metformin through gavage, and stool samples were collected at days 0, 14 and 28. All samples were then analyzed using metaproteomics. Thirty-four MS raw files were obtained with 926,004 MS/MS spectra. With an average MS/MS identification rate of 23.5% ± 12.9% (mean ± SD), 51,294 peptide sequences corresponding to 12,733 protein groups were identified with an FDR threshold of 1%. 11,870 peptides were assigned a taxonomic lineage, along with 60 genera and 79 species quantified with ≥ 3 peptides.

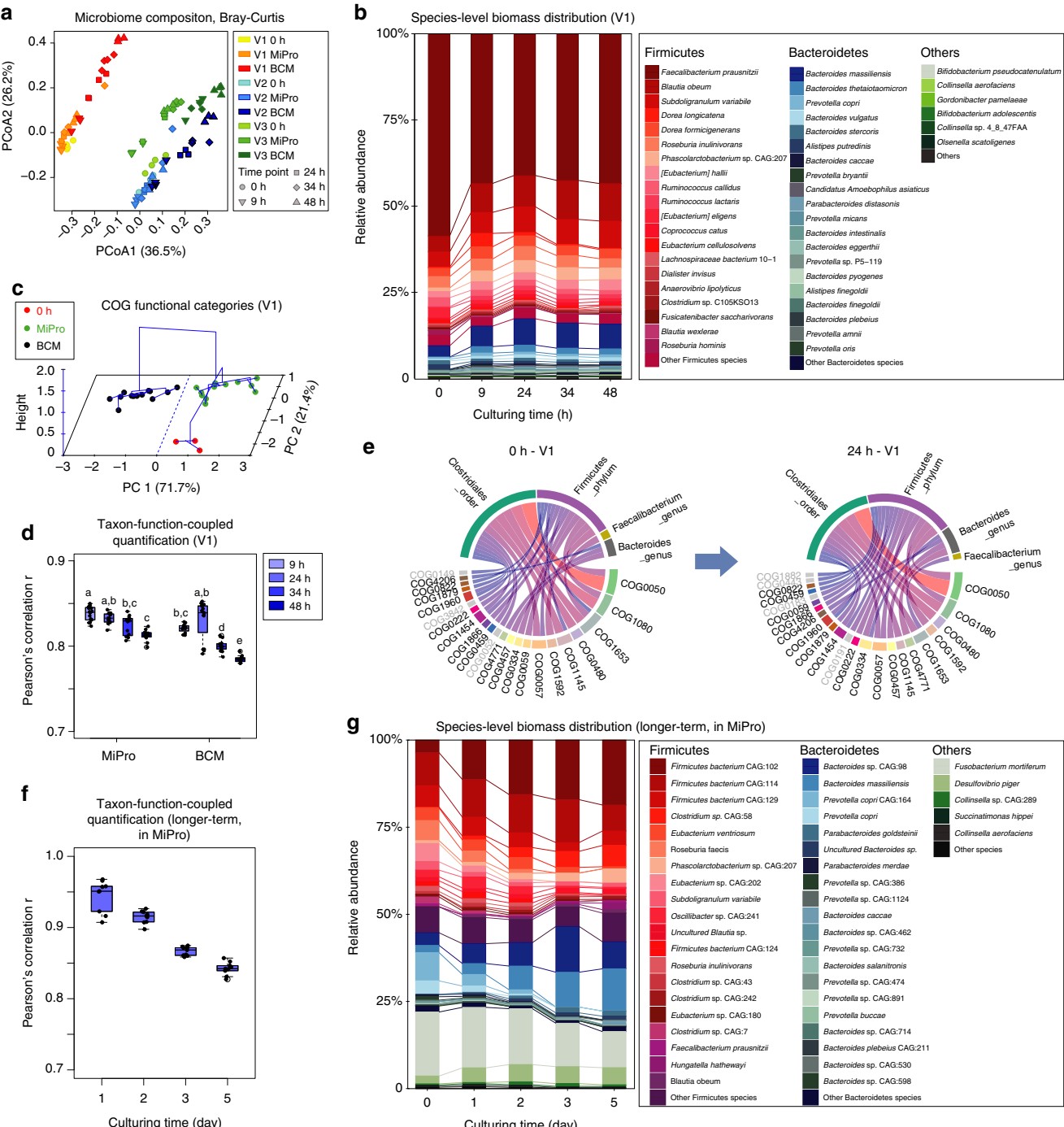

**Fig. 2** Metaproteomics revealed taxonomic & functional composition stability over time. **a** Principal coordinate analysis (PCoA) plot with Bray-Curtis dissimilarity on species level. **b** Compositional bar chart showing species-level biomass distribution over time in the cultured microbiome of V1 (see Supplementary Fig. 5 for V2 and V3). **c** PCA scores plot with hierarchical clustering based on COG functional categories of individual V1 (see Supplementary Fig. 9 for individuals V2 and V3). **d** Pearson's correlation coefficient *r* of taxon-specific functional profiles between microbiome cultured over time and the baseline inoculum sample of V1 (see Supplementary Fig. 10 for V2 and V3; different letters indicate significant differences at the *p* = 0.05 level, Tukey-b test; box spans interquartile range (25th to 75th percentile), and line within box denotes median. Whiskers represent min to max values). For 0 h, Mipro 9 h, BCM 24 h and 34 h, n = 4 technical replicates; for Mipro 24 h, 34 h and 48 h, BCM 9 h, and 48 h, n = 4 technical replicates. **e** Visualization of associated taxa-functions enrichment analysis between 24 h MiPro-cultured and 0 h baseline samples in V1 (top 30 correlations were shown). Width of blocks and linkages represents number of matched proteins. Gray letters indicate un-paired pathways within the top 30 connections. **f** Pearson's correlation coefficient *r* of taxon-specific functional profiles between the baseline inoculum and microbiome cultured in MiPro over a 5-day period. n = 3 technical replicates. **g** Compositional bar chart showing species-level biomass distribution of a gut microbiome cultured in MiPro over a 5-day period. Underlying data are provided in the Source Data file

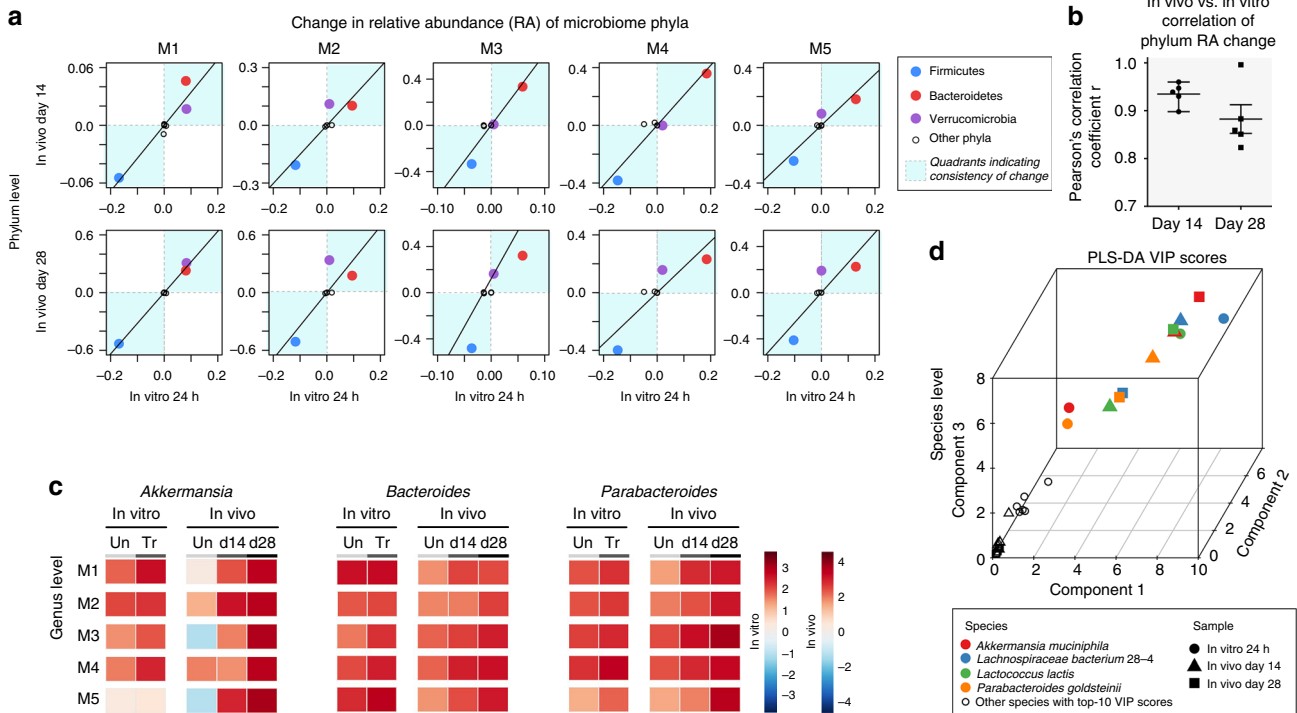

**Fig. 3** In vitro–in vivo correlation of taxonomic responses to metformin treatment. **a** Comparison of the change in relative abundance of the major gut bacterial phyla following in vitro and in vivo (days 14 and 28) metformin treatment of individual HFD-fed C57/BL6 wild-type litter-mate mice. Data point that falls into quadrants I and III (in blue background) indicates in vitro–in vivo consistency in increased and decreased phyla in response to metformin, respectively. **b** Pearson's correlation coefficient r between in vivo and in vitro changes of phyla in response to metformin treatment. Error bars represent the standard deviation. **c** Heat map showing change in relative abundance of three bacterial genera, known to increase in metformin-treated HFD-fed mice. The left panels represent in vitro cultured individual mouse microbiome in the absence or presence of metformin, whereas the right panels correspond to in vivo microbiome drug-treatment at days 14 and 28. n = 5 biologically independent mice. **d** Comparison of variable importance in projection (VIP) on the first three components from three separate PLS-DA analyses. PLS-DA analyses were performed by comparing untreated microbiome versus microbiome following in vivo 14-day, 28-day and in vitro 24 h metformin treatment, respectively. Four bacterial species were consistently ranked with the highest VIP scores in the three analyses. Underlying data are provided in the Source Data file

The IVIVCs of microbial response to metformin were evaluated for both taxon biomass contribution and functional activities. In vitro and in vivo variations in phylum-level biomass contributions were compared, as shown in Fig. 3a. Increases in Bacteroidetes and decreases in Firmicutes were observed in both in vitro and in vivo treatments (Fig. 3a). A high level of IVIVC at the phylum level, especially for day 14 samples, was indicated with Pearson's correlation of $r = 0.93 \pm 0.02$ (Fig. 3b). Furthermore, at the genus level, we observed a consistent increase in the genera *Akkermansia*, *Bacteroides* and *Parabacteroides* (Fig. 3c) under both in vitro and in vivo conditions, with the in vivo change in agreement with previously reported studies[5,42,43]. At the species level, three separate groups of partial least squares discriminant analyses (PLS-DA) were performed by comparing untreated microbiome to the in vivo microbiome after 14- and 28-days of metformin treatment and to the in vitro microbiome after 24 h of treatment (Fig. 3d). The biomass contribution of four bacterial species were consistently ranked with the highest variable importance in projection (VIP) scores (VIP > 2 for all components) across all comparisons, and a 75% agreement in these changes was observed at the species level. Metformin treatment increased the abundance of *A. municiphila*, which was in agreement with several previous studies[43–46].

For functional analysis, protein groups were first filtered with the criteria that the protein groups should be present in ≥50% in each of the listed subgroups (including in vitro untreated versus in vitro 24 h treated, in vivo untreated versus in vivo day 14 or day 28 treated samples). PLS-DA was performed on shared proteins, and proteins with a VIP score >1 were regarded as differentially expressed protein groups, which were mainly involved in 12 KEGG pathways (Fig. 4a). A total of 11,222 (out of 17,646) proteins that correspond to these KEGG pathways were extracted from the original protein group file. All LFQ intensities of protein groups that were assigned to each of the 12 pathways were summed in each sample for a KEGG pathway-level evaluation. Figure 4a shows that the relative abundances of selected pathways were uniformly altered in both in vitro and in vivo metformin-treated samples in comparison to untreated microbiomes. Subsequently, IVIVC were visualized by comparing the changed value of these KEGG pathways (Fig. 4b). In most cases, changed proteins appeared in quadrants I and III (Fig. 4b), suggesting agreement between in vitro and in vivo responses. Among these, glycolysis/glucogenesis, ABC transporters and two-component system showed high levels of variation. Here we explored further the shift of functional balance by normalizing the intensities of each enzyme against the summed protein intensity of the glycolysis/gluconeogenesis pathway (Fig. 4c). In general, the proportions of enzymes were similar in vitro and in vivo, suggesting that the in vitro model retained the functionality of the gut microbiome. Nine out of the 17 enzymes were significantly affected by metformin treatment in vivo; for the in vitro model, five significant changes were found. All other enzymes in the glycolysis/gluconeogenesis pathway showed similar trends in vitro versus in vivo, although they did not reach statistical significance.

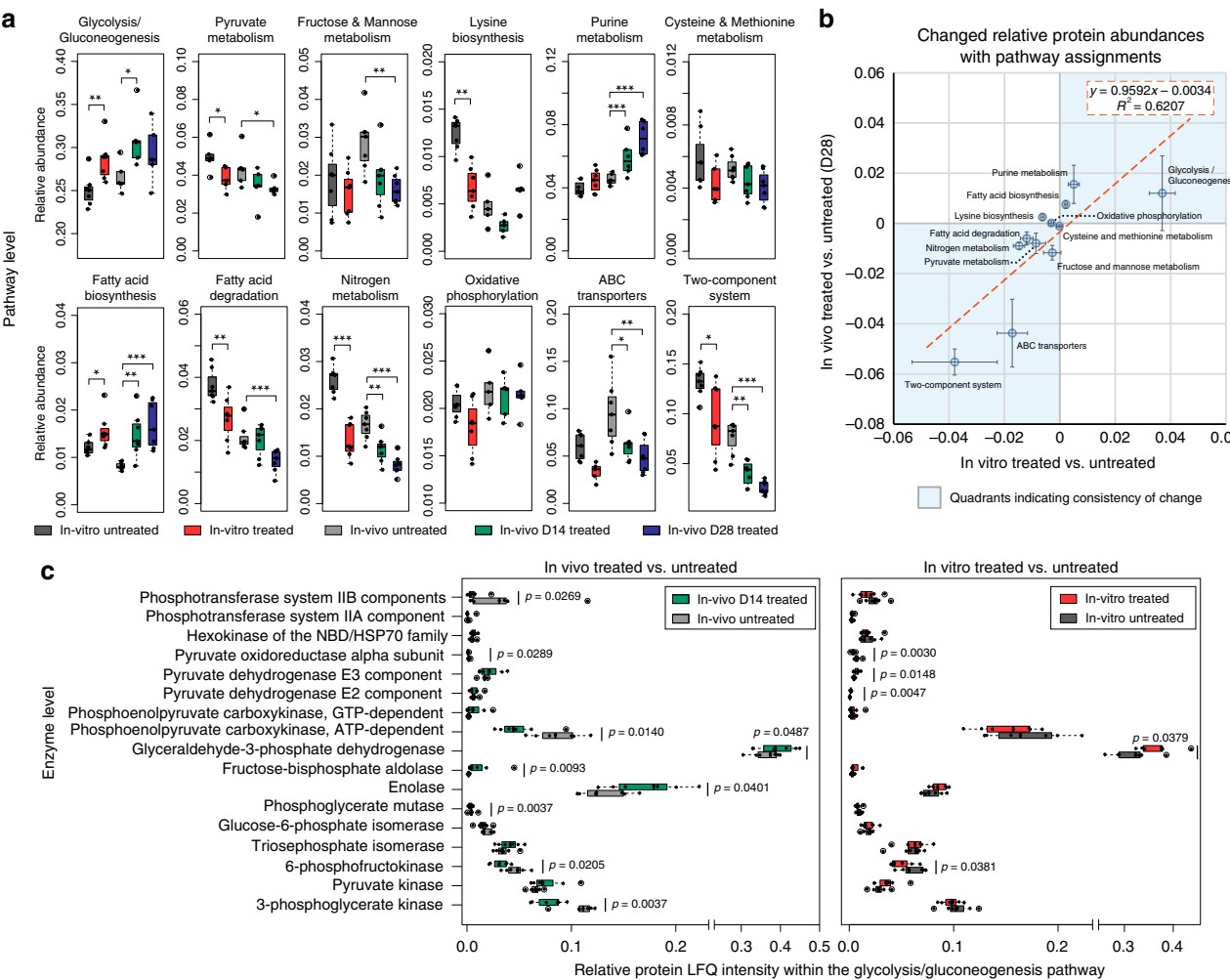

**Fig. 4** In vitro–in vivo correlation of microbiome pathway and enzyme responses to metformin. **a** Pathway analysis using the relative abundance of protein groups that are assigned to the selected KEGG pathways (KEGG pathways corresponding to PLS-DA VIP >1 protein groups). **b** In vitro–in vivo correlation of the changes in relative protein abundance with pathway assignments in microbiome samples following metformin treatment from HFD-fed mice (data shown as mean ± SEM). **c** Detailed view of the abundance of multiple enzymes in the glycolysis/gluconeogenesis pathway. Data shown here represent the relative abundance of each enzyme normalized against the total intensity of all enzymes detected in the pathway. Levels of significance are according to two sided, non-parametric $t$-test: $*p < 0.05$, $**p < 0.01$, $***p < 0.001$. Box spans interquartile range (25th to 75th percentile), and line within box denotes median. Whiskers represent min to max values. $n = 7$ biologically independent mice. Underlying data are provided in the Source Data file

## Discussion

The gut microbiome constitutes a complex microbial ecosystem in which each microbe may play a different functional role[47]. Environmental changes can trigger structural and functional alterations in the microbiome. Although some studies have shown that the gut bacterial community can be cultured in vitro[10–12,48], these studies have not demonstrated the maintenance of the microbiota's functional activities. While taxonomic profiling is important to characterize an individual's microbiome, biological processes and functions cannot be assessed by compositional analyses. Yet, such information is essential for gaining deeper understanding of the microbiota's behavior and functional activities and ultimately for designing microbiome-targeted manipulations. The need for functional understanding is even more crucial given that the microbiome's functions can be altered in absence of compositional changes. For example, sialylated milk oligosaccharides elicit a microbiota-dependent growth in mice, which are used as models of infant undernutrition by inducing the production of microbial metabolites, yet without any significant change in the relative abundance of most microbes[49]. Elucidating the microbiome's functions requires a comprehensive quantification of the transcripts,

proteins and/or metabolites produced by the microbial community. Here, we used a metaproteomic approach we recently developed[39] to simultaneously characterize the microbiome's taxonomic biomass contributions and functional activity while developing and validating a scalable in vitro model for the Maintenance of gut microbiome Profiles (MiPro).

In order to build an in vitro model to assess the microbiome response to xenobiotics, it is critical to preserve both the compositional and functional profiles of an individual's microbiome. The majority of gut microbiome culture media (including BCM)[11,12,30–36] contain a mixture of commercialized bile salts consisting of the sodium salt forms of CA and DCA at a 1:1 ratio (w/w). DCA (secondary bile acid) is known to act as an antimicrobial agent due to its high-hydrophobic and detergent effects on bacterial membranes[50]. Studies have noted a decrease in Firmicutes abundance in response to DCA[51]. Hence, we replaced DCA with CDCA, a major primary bile acid produced in human[52]. This replacement was effective in maintaining the in vitro microbiome composition (Fig. 2a). We used a silicone-gel cover perforated at the top of each to permit the escape of the gas produced by the gut microbiota while minimizing inward gas

diffusion from the anaerobic chamber. We found that using culture tubes with loose caps resulted in marked microbiome composition changes, while the 96-well format maintained the microbiome composition (Fig. 2b). One possible explanation could be that the silicon-gel cover may have increased the partial pressure of gases produced by the gut microbiome, such as $H_2$, $CH_4$, $H_2S$, and $NO_x$, etc.[53] and volatile organic compounds, such as SCFAs[54]. This could preserve the levels of dissolved gas in the culture medium. Some dissolved metabolites are important factors for bacteria cross-feeding[54] and the control of pathogens[55], and thus, are presumably essential for the maintenance of an in vitro microbiome. Other factors may include differences in the surface area to volume ratio, agitation mechanics from cultures in differently-shaped vessels. Our model was developed for rapidly screening drugs against different individual's gut microbiomes[56]. In addition, our MiPro medium may be adoptable in previously-reported fluidic-based models[18–20,57]. By incorporating the medium into gut-on-a-chip systems[20] or similar models, we may be able to study organ site-specific effects. Furthermore, our model could also be used with stable isotopes labeling[39,58] for probing metabolic activity in individual microbiomes.

After our optimizations, our MiPro model quintupled viable bacteria count, and had alpha-diversity and microbial composition at 24 h post-inoculation that were similar to the inoculum at time 0. These findings are in agreement with the premise that high biodiversity may enhance the temporal stability of microbial communities[59]. In terms of functions, a general view of functional stability using summed protein abundances in each functional category is not sufficient due to the complexity of functional constitution in a gut microbiome ecosystem. At the microbiome level, functional pathways are stable within healthy human populations[60]. This stability is due to functional redundancy among species in a gut microbiome[29]. Indeed, functional compensation can happen among different taxa, which preserves long-term average ecosystem performance, as one species can increase in function to balance for loss or decline in that function by another species[61]. This mechanism shifts the functional balance between two taxa in a gut ecosystem. Therefore, for maintenance of an in vitro gut microbiome culturing system, it is very important to preserve the stability of taxon-specific functional profiles in the microbiome. And results showed that this has been well achieved in our model. Together, our MiPro model comprises two phases: a growth phase with significant viable biomass accumulation and a lag phase where >83% of the taxon-specific functional activities of the inoculum were maintained. Hence, this model can be effectively used to amplify a microbial community and to investigate the effect of perturbations on the microbial community composition and functional activities. As stated previously, the gut microbiome can respond to changes in its environment within a day[21–23]. Hence, for processes such as high-throughput drug screening, a 24 h culture using the Mipro model would be applicable. Moreover, in consideration of exceptions such as xenobiotics that may act slowly, we verified that the MiPro model could sustain gut microbiome functionality for up to five days.

The IVIVC study estimated the power of the MiPro to recapitulate in vivo drug responses at different levels of taxonomic biomass contributions and functional activities. Changes of major taxonomic responders on phylum, genus and species levels as reported in several studies[5,42–46], were captured in our subsequent metaproteomic analyses from both our in vitro and in vivo experiments. Additionally, our functional profiling analysis showed agreements between our in vitro and in vivo responses, and as well as, with previous studies[5]. Significant responses to metformin were found in several pathways: glycolysis/gluconegenesis, pyruvate metabolism, fatty acid biosynthesis,

fatty acid degradation, nitrogen metabolism, ABC transporters, and the two-component system. A recent study[5] indicated that metformin alters multiple microbial pathways including ABC transporters, two-component system, fructose and mannose metabolism and pyruvate metabolism. All of these responses were found in our model. Two microbial ABC transporters and the two-component systems were significantly decreased, while the glycolysis/ gluconeogenesis pathway was increased, following drug treatments in both the MiPro and in vivo models (Fig. 4a). Shin et al. reported that metformin improves glucose homeostasis in HFD-fed mice[43]. Changes in the glycolysis/gluconeogenesis pathway within the gut microbial environment could play a key role in glucose absorption across the intestinal mucosal layer[62]. Notably, metaproteomic responses within the glycolysis/gluconeogenesis pathway revealed high IVIVC of the MiPro at the enzyme level (Fig. 4c). This highlighted the depth of our model to recapitulate in vivo microbiome functional activities in response to drug treatment.

Although functional activities of the gut microbiota were determined using metaproteomics, microbiome metabolites generated throughout the culturing process were not evaluated in this study. Accumulation of secondary metabolites could be a factor that gradually alters microbiome functionality. For example, in our study we showed that adding a bacterial-metabolite, namely a secondary bile salt, induced a shift in the microbiota. Therefore, future work will explore whether the formation of secondary and tertiary bile acids during culturing affects the microbiome. Although the composition and functions were maintained for each individual microbiome, our study was not designed to detect changes in very low abundance bacterial species. In addition, factors such as different protein extraction protocols and bioinformatic database may introduce sources of bias in metaproteomic analysis[63]. Further in-depth investigations on in vitro culturing and drug responses could combine metaproteomics with other omics technologies such as metagenomics and metabonomics to gain deeper insights of the microbiome structure and function.

In conclusion, we evaluated the ability of the MiPro model to maintain microbial taxon-function stability and tested the utility of this model for high-throughput drug-microbiome interactions studies. We optimized the medium and culture model for scalable microbiome culturing. The optimized model showed improved performance in sustaining viability, diversity, compositional and functional profiles of the inoculum microbiome. We observed a high degree of in vitro–in vivo correlation of compositional and functional responses to metformin treatment. Our work provides an effective experimental platform for drug-microbiome interaction studies.

## Methods

**Medium preparation and gut microbiome culturing**. The medium composition for MiPro was based on our previously suggested medium composition[22], which comprises: 2.0 g $L^{-1}$ peptone water, 2.0 g $L^{-1}$ yeast extract, 0.5 g $L^{-1}$ L-cysteine hydrochloride, 2 mL $L^{-1}$ Tween 80, 5 mg $L^{-1}$ hemin, 10 µL $L^{-1}$ vitamin K1, 1.0 g $L^{-1}$ NaCl, 0.4 g $L^{-1}$ $K_2HPO_4$, 0.4 g $L^{-1}$ $KH_2PO_4$, 0.1 g $L^{-1}$ $MgSO_4\cdot7H_2O$, 0.1 g $L^{-1}$ $CaCl_2\cdot2H_2O$, 4.0 g $L^{-1}$ $NaHCO_3$, 4.0 g $L^{-1}$ porcine gastric mucin (cat# M1778, Sigma-Aldrich), and 0.5 g $L^{-1}$ bile salts. Based on this composition, two media containing either commercialized mixed bile salts (0.5 g $L^{-1}$, 1:1 sodium cholate: sodium deoxycholate mixture, cat# 48305, Sigma-Aldrich) or a primary bile salts composition (0.25 g $L^{-1}$ sodium cholate and 0.25 g $L^{-1}$ sodium chenodeoxycholate) were compared for further optimization. To identify the ideal physical culturing conditions, samples were cultured either in 1 ml of media in 96-deep well plates or in 2 ml of media in culturing tubes (cat#T406-2A, Simport, Canada). The 96-deep well plate was covered with a silicone gel mat with a vent hole on each well made by a sterile syringe needle. During culturing, plates and tubes were shaken at 500 rpm with digital shakers (MS3, IKA, Germany). The optimal medium and culturing condition for the MiPro model was then determined based on its ability to maintain gut microbiome composition assessed as described below.

The human stool sampling protocol was approved by the Ottawa Health Science Network Research Ethics Board at the Ottawa Hospital (#20160585-01H).

All participants signed informed consent to participate in the study. Briefly, ~ 3 g of fresh stool sample was collected from each individual using a 2.5 ml sterile sampling spoon (Bel-Art, United States). The spoon was dropped into a 50 ml Falcon tube containing 15 ml of sterile PBS pre-reduced with 0.1% (w/v) L-cysteine hydrochloride. The samples were immediately transferred into an anaerobic workstation (5% $H_2$, 5% $CO_2$, and 90% $N_2$ at 37 °C). Before homogenization with a vortex mixer, the tube was uncapped for a few seconds to enable gas exchange and in particular oxygen removal. Sample homogenates were filtered using sterile gauzes and were immediately inoculated into each medium for static culturing at a final inoculum concentration of 2% (w/v). Culturing of an individual's microbiome was carried out in 1 ml MiPro medium. The conventional basal culture medium (BCM)[11] was also included for evaluating the performance of our improved medium. For each individual, 32 replicates were cultured for each medium, allowing for 4 replicates at 8 different time points. Microbiomes were characterized at 0 (immediately after inoculation), 3, 6, 9, 12, 24, 34, and 48 h by measurements of the optical density at 595 nm ($OD_{595}$), as a proxy of microbial growth and biomass, and by metaproteomic analyses. For testing the performance of MiPro over a longer period of time, an individual gut microbiome was cultured in the MiPro model, and 250 μl of the cultured suspension was replaced with same volume of fresh medium every 12 h. Microbiomes were collected at 0 h, 1-, 2-, 3-, and 5- days for metaproteomic analysis.

**Microbiome growth and viability tests**. At 0, 3, 6, 9, 12, 24, 34 and 48 h, two 100 μl aliquots were removed from each sample for $OD_{595}$ measurements. One of the aliquots was centrifuged at 16,000 ×g and the supernatant was used as the medium blank for the $OD_{595}$ measurement. Viability of the microbiomes was tested at 0, 9, 24, and 48 h using the LIVE/DEAD BacLight Kit (Thermo Fisher Scientific Cat# L7012) in combination with flow cytometry[64]. Briefly, according to the manufacturer's instruction, microbial cells were washed and diluted in 0.85% NaCl saline buffer. Then, the bacteria were stained with PI and SYTO for 15 min prior to acquisition on a BD FACSCelesta™ multicolor cell analyzer. For maximum bacterial cell viability, all sample processing procedures, including mixing, dilution and staining steps were performed inside the anaerobic station. For each sample, data were recorded for 2 min on the low flow rate setting. Bacterial cells were gated according to size and granularity on the FSC/SSC scatter plot. Live/dead cells were gated according to the gates set by the stained gut microbiome, the heat-treated and stained gut microbiome, and the unstained gut microbiome (Supplementary Fig. 3). Data were analyzed using Kaluza Analysis Software version 1.5. Live-gated bacterial cell counts were compared for its fold change in the cultured microbiome with respect to the inoculum, i.e. (cultured—inoculum)/inoculum.

**MiPro model to in vivo comparison and validation**. In vitro and in vivo effects of metformin on the murine gut microbiome associated with a high-fat diet (HFD) were tested. Briefly, 7-week old male litter-mates from inbred C57/BL6 mice were single-housed and fed a 42% fat (by calories) diet (ENVIGO, TD.09682) for 6 weeks to allow stabilization of their microbiomes. Fresh stool pellets from each mouse were collected for in vitro culturing on day-0. The microbiomes were cultured in the absence or presence of metformin using the MiPro model. The in vitro concentration of metformin was set to 6 mg/ml, emulating the 30% fecal recovery ratio of metformin previously reported from in vivo experiments[41]. Cultured microbiome samples were harvested at 24 h for metaproteomic analysis. For the in vivo experiment, mice were treated daily with 300 mg/kg of metformin through oral gavage. Stool pellets were collected for metaproteomic analysis at days 0, 14 and 28. The animal experiment was performed at the University of Ottawa and conducted in strict accordance with the guidelines of the Care and Use of Experimental Animals of Canadian Council on Animal Care (CCAC). The animal use protocol was approved by the Animal Care Committee at the University of Ottawa (# BMI 2848).

**Trypsin digestion, desalting, and LC-MS/MS analysis**. Cultured gut microbiome samples were pelleted at 14,000 × g, 4 °C for 20 min, followed by two rounds of cell washing with PBS. Then large debris were removed with 300 × g centrifugation at 4 °C for 5 min, followed by pelleting cells at 14,000 × g, 4 °C for 20 min. The microbial cell pellets were lysed by ultrasonication (Q125 Sonicator, Qsonica, LLC) in 200 μL lysis buffer (4% (w/v) sodium dodecyl sulfate and 8 M urea in 50 mM Tris-HCl buffer, pH 8.0; plus Roche PhosSTOP™ and Roche cOmplete™ Mini tablets), and proteins precipitated overnight in acidified acetone/ethanol at −20 °C. Proteins were washed three times with ice-cold acetone, then dissolved in 6 M urea in 50 mM ammonium bicarbonate (pH = 8). Protein concentrations were determined by DC (detergent compatible) protein assay before an overnight trypsin digestion at 37 °C following protein reduction and alkylation 10 mM dithiothreitol and 20 mM iodoacetamide, respectively. Desalted tryptic peptides corresponding to 1 μg of protein were loaded for LC-MS/MS analysis with an Agilent 1100 Capillary LC system (Agilent Technologies, San Jose, CA) and run on a Q Exactive mass spectrometer (ThermoFisher Scientific Inc.). Peptides were separated with a tip column (75 μm i.d. × 50 cm) packed with reverse phase beads (1.9 μm/120 Å ReproSil-Pur C18 resin, Dr. Maisch GmbH, Ammerbuch, Germany). Peptide separation was performed using either a 90 min gradient for human samples or a 120 min gradient for mouse samples. The gradients included 5 to 30%

(v/v) acetonitrile at a flow rate of 200 nL/min, for which 0.1% (v/v) formic acid in water was used as solvent A, and 0.1% FA in 80% acetonitrile was used as solvent B. The instrument parameters included a full MS scan from 300 to 1800 $m/z$, followed by data-dependent MS/MS scan of the 12 most intense ions, a dynamic exclusion repeat count of two, and repeat exclusion duration of 30 s. All samples were run on LC-MS/MS in a random order. In addition, for the pre-experiment that evaluated bile salts composition and culture conditions, samples were analyzed on an Orbitrap XL following a 6 h gradient of 5–25% acetonitrile (v/v) at a flow rate of 300 nL/min, with 0.1% formic acid (FA) in water as solvent A and 0.1% FA in acetonitrile as solvent B. The full MS scan range was 350–1800 $m/z$, followed by five MS/MS scans performed in ion trap by CID (collision energy, 35%) in positive mode, dynamic exclusion repeat count of one, repeat duration of 30 s, and exclusion duration of 90 s.

**Metaproteomics data processing**. Protein/peptide identification and quantification were carried out using the MetaLab software (version 1.0)[37], which automates the MetaPro-IQ approach[39]. Database construction were based on an iterative database search strategy using gut microbial gene catalogs (for cultured human microbiomes, human gut microbial gene catalog with 9,878,647 sequences from http://meta.genomics.cn/ and for mice microbiomes, mouse gut microbial gene catalog database comprising 2,572,074 genes, obtained from http://gigadb.org) and a spectral clustering strategy. The peptide and protein lists were generated by applying strict filtering based on a FDR of 0.01, and quantitative information of proteins were obtained with the maxLFQ algorithm; quantitative taxonomic analyses were achieved by assigning identified peptides the taxonomic lineage of their lowest common ancestor (LCA) through the pep2tax database and summing up the intensities of the peptides. Functional annotations (COG, KEGG) were obtained with MetaLab version 1.1.0.

For human gut microbiomes, protein groups were filtered with the criteria that the protein should be present in ≥25% of the samples (Q25). Bray-Curtis dissimilarity and analysis of similarities (ANOSIM) were performed using the R package vegan. Principal coordinates analysis (PCoA), principle component analysis (PCA), and hierarchical clustering were visualized with R (version 3.4.3). Taxonomic composition visualization and taxon-function coupled analysis were performed and visualized using iMetaLab (http://imetalab.ca/). The database of clusters of orthologous groups (COG) of proteins was used for functional annotation. For each sample, taxon-specific functional proteins with protein intensity was then generated from iMetaLab (Supplementary Fig. 1). With these sets of tables, the Pearson's correlation coefficient $r$ of the taxon-function coupled profile between any two samples was calculated using R function cor() to generate a correlation matrix, and the correlation between 0 h and the cultured microbiome at each subsequent time point was obtained. For visualizing the taxon-function coupled enrichment, $p < 0.05$ was set as the threshold for both taxonomic and functional enrichment, and the top 30 connections were selected from enriched taxon-function matches.

For the in vitro–in vivo microbiome experiments, protein groups were filtered with the criteria that the protein groups should be present in ≥50% (Q50) in each of the listed subgroups (including in vitro untreated versus in vitro 24 h treated, in vivo untreated versus in vivo day 14 or day 28 treated samples). A partial least squares—discriminant analyses (PLS-DA) test was performed on shared proteins among listed subgroups using the online tool MetaboAnalyst (www.metaboanalyst.ca/). Protein groups with a VIP score >1 were annotated to the corresponding KEGG categories. Then all protein groups annotated with these KEGG categories were extracted from the original protein group file. In vitro–in vivo correlation of the microbiome drug response was evaluated based on taxonomic and pathway changes after metformin treatment. Pearson's correlation coefficient $r$ was calculated using R function cor(). Taxonomic composition analysis was done using MetaboAnalyst.

**Reporting summary**. Further information on research design is available in the Nature Research Reporting Summary linked to this article.

## Data availability

All raw data from LC-MS/MS have been deposited to the ProteomeXchange Consortium (http://www.proteomexchange.org) via the PRIDE partner repository (dataset identifiers PXD010134, PXD010135 and PXD013600). The source data underlying Figs. 1b, c, 2d, f, 3b and 4a–c and Supplementary Figs. 7, 8 and 10a, b, are provided as a Source Data file.

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

## Acknowledgements

This work was supported by the Government of Canada through Genome Canada and the Ontario Genomics Institute (OGI-114), CIHR grant (ECD-144627), the Natural Sciences and Engineering Research Council of Canada (NSERC, grant no. 210034), and the Ontario Ministry of Economic Development and Innovation (REG1-4450). This work was also funded by the Government of Canada through Genome Canada and the Ontario Genomics Institute (OGI-156), as well as funding from the Province of Ontario through the Ontario Research Fund (DIG-14405). The authors thank Dr. James Butcher and Jennifer Li for the 16S rDNA sequencing contributing to the reviewing process, and thank Dr. Kendra Hodgkinson for proofreading.

## Author contributions

L.L., E.A.S., and D.F. designed the study. D.F. supervised the study. L.L., E.A.S., J.M., Z.N., J.W., and K.W. performed the experiments. L.L., E.A.S., Z.N., X.Z., and K.C. performed data analysis. L.L., E.A.S., and D.F. wrote the paper. A.S., D.F., X.Z., J.M., and Z.N. contributed to the editing and revision of the paper. All authors read and approved the final manuscript.

## Additional information

**Competing interests:** A.S. and D.F. have co-founded Biotagenics and MedBiome, clinical microbiomics companies. The remaining authors declare no competing interests.

