## [Peer Review File · Nature Communications]

Reviewers' comments:

Reviewer #1 (Remarks to the Author):

The manuscript tackles a very important aspect in microbiome research. I am focussing in my review mainly on the metaproteomics part of the study.

Minor criticism:

1. The authors should mention all the other open questions that would benefit from a functionality conserving culturing like organ site specific effects, and application of stable isotopes for probing metabolic activity species-wise.
2. The real advantage of metaproteomics lies in the combination of functional and phylogenetic information.
3. The authors should also mention and use metabolome analysis for confirmation.
4. Was the concentration of bile acids constant over time?
5. How many secondary and tertiary bile acids were formed?
6. The authors should state somewhere how many psm were found and how many species were detected as well as a number of psm/taxa.
7. The detected 1000 COGs from 419 taxa are in the normal range of such a metaproteomics study. Interesting are two things, first the distribution over taxa and a comment how this relates to other estimations of bacterial diversity from gut samples.
8. For the functional analysis the same order should be used in figure 2 and the authors should set a minimum for the pathway coverage to about 20% in order to gain reliability.

Major criticism:

1. Although metaproteomics is indeed well suited for functional characterisation the analysis of 16srRNA is still the old gold standard. If the authors want to apply the best technique for analysing the community structure they should apply metagenomics.
2. Is it possible to sample SCFAs with this culturing approach?
3. The authors should determine the maximal run time for an experiment in this culturing approach.

Conclusion:

In-vitro culturing that restores the different complex features of a microbiota sample is really highly relevant. And the metaproteomics is a good starting point for proving that but there are several aspects missing before the paper can be published.

Reviewer #2 (Remarks to the Author):

This work has been done to demonstrate the utility of a batch culture method for maintenance of ecosystem function for study in a high-throughput manner.

The authors claim that their method of culture of the gut microbiota in deep-well 96-well plates – the basis of the MiPro – is novel and that this method of culture maintains the composition of the microbiota over time, thus mimicking the gut microbiota function better than in bioreactor systems where flow conditions are applied to allow the culture to attain steady state. I agree that a limitation of more complex in vitro models is the change that the supported ecosystem has to undergo in order to cope with its new in vitro conditions (mostly these changes are related to substrate availability). However, I do not think that the MiPro system is actually a valid comparator. It is essentially a high throughput batch culture system, however the stability that it claims to have has only been measured over a maximum of 48 hrs. During this time, the cell mass within each well increased, but this is expected. The novelty seems to come from suggesting that

the end result after just 48 hrs looks similar to the inoculum, but I think this would be true of any system where only 4 generations were followed, and I am struggling to see what is new here. The proteomics work done, while it is good and careful work, is likely not very representative of the new biomass, but of the original inoculum.

If one also considers the work of Auchtung et al. (Microbiome 2015; 3:42), there is also really no novelty in the high throughput aspect of the current work, since this has been done before and, indeed, validated across different individuals.

I am troubled by the use of OD as a measure of growth. This is a proxy for growth only, and cannot accurately be used for mixed species ecosystems where multiple species of highly variant cell size are to be considered (such as that of the gut microbiome). Since flow cytometry was carried out, why not use this to accurately count biomass?

I would like to see the results gained from tube culture of the 3 tested ecosystems presented over time and compared to the MiPro culture, as is presented for the MiPro culture in figure 2. Otherwise, only a global Firmicutes:Bacteroidetes ratio is shown in figure 1 – but this is not really enough to draw conclusions about the comparison between the 2 methods of culture.

The strength of the manuscript lies in the approach used to compare samples through metaproteomics (although not in itself a novel approach) and the testing of different bile salt mixtures on the microbial ecosystems, both of which are useful additions to the field. However, I would like to see more evidence that this MiPro model is truly unique in its ability to support ecosystems while maintaining inocula specific characteristics – to do this would require following the system for at least 72 hrs and possible even longer. Otherwise, all that this system represents is a simple, and not really all that novel, batch culture system for the gut microbiome that only really has any utility for short-term experiments and drug interaction studies.

Reviewer #3 (Remarks to the Author):

General comments:

This is a well-designed study that addresses the important issue of developing in vitro systems for culturing and testing complex bacterial populations such as the gut microbiome. Correlating the in vitro results from the authors' MiPro method with in vivo results is well done.

The Results section and the Methods section are, in general, well written for both clarity of thought and language. The Introduction and Discussion sections (and to a lesser degree the Abstract) appear to have been written by a different author and may suffer from the challenge of writing in English as a second language. Several specific examples are pointed out below, but these sections should be read and edited by the other authors to obtain a uniformly high level of presentation throughout the manuscript.

Specific points:

1. The authors describe the MiPro system as suitable for high-throughput investigations. While the method scales well for sample replication and multiple testing conditions, the length of analysis time per sample (90-120 minute gradients for peptide separation) suggests that a single 96-well plate would require more than 6 days of LC-MS/MS instrument time alone. Calling this "high-throughput" is arguably little strong.

2. On page 4 the authors state that they "observed a 4.4-fold increase of viable bacterial count in MiPro medium after 24 hr of culturing, whereas a 3.0-fold maximum increase was detected in the

BCM medium at 9 hr post-culturing.” It is not clear how these fold increases were derived from the data presented in Figure 1. A better description – either in the Results or Methods section – is necessary.

3. In the Results section on page 3, the authors suggest that the superiority of the deep well plate culturing method over culture tubes may be due to reduced gas-exchange with the outer environment and preservation of partial pressure of gasses and volatile metabolites. While it isn't immediately clear why a perforated membrane would allow less gas transfer than a loosened cap, this explanation is at least plausible. However, on page 12 in the Discussion section, the authors state the above explanation as establish fact. If you have done any gas analysis of the headspace to prove this, it isn't presented or described. There are other possible explanations for the difference (i.e. differences in surface area to volume in the cultures, agitation mechanics in differently-shaped culture vessels, or others). The statement in the Discussion should be written as a possibility – not fact.

4. Page 6: On one line the samples from the MiPro are described as “OM-cultured samples.” This was the only use of “OM-cultured” that I could find in the manuscript. Legacy from an earlier version? Either define the term or change it to be consistent with the rest of the manuscript.

5. In the legend for Figure 1, the description for panels C and D are mislabeled as B and C, respectively.

6. In Figure 2 panel E, the COGs 0149, 3842, and 0052 in the 0 hr diagram are not paired with pathways in the 24 hr diagram. Shouldn't these be in gray text as well?

7. The first two paragraphs of the Discussion section have several sentences duplicated verbatim. Choose one location or the other.

8. The phrase “time-and-cost saving solutions” (Abstract [page 1] and Introduction [page 2]) is atypical and grammatically questionable.

9. There are multiple misspellings or incorrect word choices scattered throughout the manuscript (e.g. equal should be equate [page 2], alternations should be alterations [page 11], health should be healthy and recapitulate should be recapitulate or replicate [page 13], etc.). Careful reading and editing is needed throughout.

The authors present interesting and useful results, particularly with respect to the bile acid composition of the growth medium, the effect of different culture vessels, and the ability of the assay to mirror in vivo results under the conditions they tested. There are some potential limitations to the MiPro approach, however. For example, the use of a dilute initial bacterial concentration and dependence of the method on bacterial growth in a 24 hour period raises the possibility of masking the contributions of bacterial species with slower growth rates. Also the 24 hour window may be too short for some types of drug or xenobiotic exposure studies. Using a metaproteomic approach to the analysis to generate both biomass and function data is efficient, but does it introduce biases in bacterial identification that could create problems? The Discussion and Conclusion sections tout the benefits of the method, but some additional discussion of the potential limitations should be included as well.

Point-to-point response to reviewers' comments

Reviewer #1 (Remarks to the Author):

The manuscript tackles a very important aspect in microbiome research. I am focusing in my review mainly on the metaproteomics part of the study.

Minor criticism:

1. The authors should mention all the other open questions that would benefit from a functionality conserving culturing like organ site specific effects, and application of stable isotopes for probing metabolic activity species-wise.

Response: We would like to thank the reviewer for the suggestions to improve our manuscript. We have added a discussion about these points in the Discussions (page 13, lines 317-320), marked in blue.

“By incorporating the medium into gut-on-a-chip systems²¹ or similar models, we may be able to study organ site-specific effects. Furthermore, our model could also be used with stable isotopes labeling^{40,59} for probing metabolic activity in individual microbiomes.”

2. The real advantage of metaproteomics lies in the combination of functional and phylogenic information.

Response: We have added this point in the Introduction, page 2, lines 49-51, marked in blue.

“An advantage of metaproteomics when compared to sequencing-based techniques is that it can combine functional and phylogenic information.”

3. The authors should also mention and use metabolome analysis for confirmation.

Response: We thank Reviewer #1 for the constructive suggestion. The primary goal of our manuscript is to assess the functional changes of the microbiome in addition to phylogenic composition. Functional changes can be assessed using metaproteomics.

We agree with the reviewer that the next step will be to examine the metabolome. However, a metabolomics analysis is beyond the scope of our current manuscript as it will take another 1-2 years to optimize and validate the experimental protocols to allow reliable metabolites

identification and quantification. As this represents a limitation of our current study, we have added the following sentence acknowledging this fact to our discussion, page 15, lines 360-365, as follows:

“ Although functional activities of the gut microbiota were determined using metaproteomics, microbiome metabolites generated throughout the culturing process were not evaluated in this study. Accumulation of secondary metabolites could be a factor that gradually alters microbiome functionality. For example, in our study we showed that adding a bacterial-metabolite, namely a secondary bile salt, induced a shift in the microbiota. Therefore, future work will explore whether the formation of secondary and tertiary bile acids during culturing affects the microbiome.”

References:

- L Li, Z Ning, X Zhang, *et al.* (2019), RapidAIM: A culture- and metaproteomics-based Rapid Assay of Individual Microbiome responses to drugs, bioRxiv, doi: 10.1101/543256

4. Was the concentration of bile acids constant over time?

Response: The reviewer raises an important point because we found that adding different compositions of bile acids in the culture medium can affect the maintenance of the *in vitro* microbiome. Gut bacteria can convert primary bile acids to secondary bile acids, so over time we would predict an accumulation of secondary bile acids that may affect the gut microbiome. In this revised version, we did an additional experiment to culture the gut microbiome for five days. We examined bile acid metabolism pathway-related proteins (COG1028 function) and observed stable protein levels over five days of culture (Figure R1). Nevertheless, it will be very interesting to focus on bile acids as a separate follow-up study, using metabonomics that will measure bile acids directly.

Figure R1. Bile acid synthesis pathway-related proteins during the five-day culturing

5. How many secondary and tertiary bile acids were formed?

Response: As discussed above (Comment 4), we have not yet optimized our protocols for the metabolomics analysis, which will allow us to measure bile acids. As our manuscript suggested that bile acid composition affects microbiome response, we agree with the reviewer that it would be very interesting to focus on bile acids in a separate follow-up study. We have added the following sentence acknowledging this fact to our discussion, page 15, lines 360-365 (same lines as in comment 3).

6. The authors should state somewhere how many psm were found and how many species were detected as well as a number of psm/taxa.

Response: We are sorry about missing this information. Using MetaLab searching (with MaxQuant software embedded), the PSM is shown as “MS/MS Identified” in the result table, and subsequently an “MS/MS identification rate” is given by calculating “MS/MS Identified” over “MS/MS submitted”. We had mentioned “MS/MS submitted” in the previous manuscript, and in the revisions we added the “MS/MS identification rate” (page 6, lines 130-131; page 9, lines 20-21). For the number of taxonomic identifications, we had mentioned the number of identified peptides that were matched to taxa in the *in vitro* study

section but missed this information in the *in vivo* validation section. We have added this information in the results (page 9, lines 216-219).

Page 6, lines 130-131 (in vitro culturing study, human microbiome): “With an average MS/MS identification rate of $40.8\% \pm 4.5\%$ (mean \pm SD), a total of 58,848 peptides and 16,326 protein groups were identified with a false discovery rate (FDR) threshold of 1%.”

Page 9, lines 216-219 (in vivo - in vitro correlation, mouse microbiome): “With an average MS/MS identification rate of $23.5\% \pm 12.9\%$ (mean \pm SD), 51,294 peptide sequences corresponding to 12,733 protein groups were identified with an FDR threshold of 1%. 11,870 peptides were assigned a taxonomic lineage, along with 60 genera and 79 species quantified with ≥ 3 peptides.”

7. The detected 1000 COGs from 419 taxa are in the normal range of such a metaproteomics study. Interesting are two things, first the distribution over taxa and a comment how this relates to other estimations of bacterial diversity from gut samples.

Response: Thank you for the thoughtful comment. We have visualized the functional distribution over taxa and added it as a supplementary figure (Figure S1A, as shown below). This figure was generated using COG-taxa matching results combining all data on iMetaLab.ca. To ensure proper visibility, we set a cut-off of “number of valid values for each row” ≥ 5 and “number of valid values for each column” ≥ 10 in the figure. We have also added a figure of the three-dimensional dataset (sample-taxon-function) in Supplementary Figure S1B (as shown below) and some descriptions in the text (page 7, lines 163-166), as follows:

“In total we identified 1,066 unique COGs of proteins corresponding to 419 taxa, the overall taxon-function distribution across the dataset is shown in Supplementary Figure S1A. By generating the taxon-function distribution for individual samples, a three-dimensional dataset for between-sample comparisons was created (sample-taxon-function, Supplementary Figure S1B).”

For taxonomic analysis such as beta-diversity, we used peptides to quantify taxonomic biomass contribution through a pep2tax database; and for taxon-specific functional analysis we used protein LFQ intensity generated by maxLFQ in Maxquant based on the peptides. The results were related based on the same set of quantified peptides, just that there was a conversion to protein groups before the taxon-specific functional analysis.

Figure R2 (Figure S1 in manuscript). Taxon-specific functional activity. Taxon-function-coupled analysis was carried out using the iMetaLab platform (<http://shiny.imetalab.ca/>) using the enrichment analysis module. By setting the enrichment p value threshold to 1, all COGs corresponding to all taxa were obtained without any filtering. (A) An overview of taxon-specific functional activity distribution in the *in vitro* validation dataset (three individual gut microbiomes inoculated 0-48 hrs in MiPro and BCM media) by selecting the “combined all samples”. (B) By selecting individual sample labels, the taxon-specific functional activity in each sample was obtained, generating a three-dimensional dataset (sample-taxon-function) for comparisons.

8. For the functional analysis the same order should be used in figure 2 and the authors should set a minimum for the pathway coverage to about 20% in order to gain reliability.

Response: Thank you for the suggestion. For the functional analysis in Figure 2, first we'd like to clarify that Figure 2D uses 100% pathway coverage of our data, so we believe that the result has sufficient reliability. As for Figure 2E, we understand Reviewer #1's concern that we only showed the top 30 pairs of enriched taxon-functions, visualizing a low coverage of the pathways. For each sample, we roughly obtained 3000 pairs of enriched taxon-functions ($p < 0.05$). Unfortunately, that is very difficult to display in a sub-panel. Therefore, we have also added a Supplementary Figure S10 with the top 300 enriched taxon-function correlations (Figure R3). The order of COGs is generated automated according to the number of correlations in each node, and it's difficult to directly compare from the circus plot with the high number of features, so we added a panel C of Venn plot to compare between the figure

panels A and B. It shows highly overlapped taxon and function enrichment between the inoculum and 24 hr cultured microbiome.

Figure R3 (Figure S10 in manuscript). Comparison of top 300 enriched taxa-function correlations, between (A) 0 hr baseline sample and (B) 24 hr MiPro-cultured sample; (C) Venn diagrams showing overlapped taxa and functions between the two groups.

Major criticism:

1. Although metaproteomics is indeed well suited for functional characterization the analysis of 16srRNA is still the old gold standard. If the authors want to apply the best technique for analyzing the community structure they should apply metagenomics.

Response: Metagenomics enables deep taxonomic resolution and provides information about the potential microbiome functions. However, metagenomics offers limited insight on which microbial traits actually contribute to the functional activities of the microbiome, as genes predicted from metagenomics analyses are not necessarily expressed. In contrast, in addition to precisely quantifying microbial functional proteins that are actually expressed, MS-based metaproteomics can accurately quantify taxon-specific biomass contributions and taxon-specific functional profiles; just as Reviewer #1 have mentioned in Comment 2, the advantage of metaproteomics lies in the combination of functional and phygenic information. Nevertheless, we agree that 16S rDNA sequencing is a gold standard to analyze the community structure at the composition level. Therefore, we have validated our findings with 16S rDNA sequencing on aliquots of the samples that have been kept as backup from the *in vitro* culturing study. We analyzed the microbiome composition at family, genus and species levels, as shown below. The first column in each figure is the 0 hr inoculum, and second and third columns are 24 hr and 48 hr culture in MiPro; the forth column is 24 hr culture in BCM medium. The results supported our metaproteomic data, confirming that the microbiome composition was reasonably at 24 and 48 hours of culturing in MiPro, and that our medium out-performed the BCM medium.

Figure R4. 16S rDNA sequencing showing stability of microbiota in the MiPro medium

2. Is it possible to sample SCFAs with this culturing approach?

Response: Yes, it is feasible to extract and analyze short chain fatty acids from *in vitro* cultured gut microbiota. Several methods have been used: for example, in a study by Alander *et al.* (1999), SCFAs were extracted from gut microbiota cultured in the SHIME system using diethylether and were analyzed through GC. Costabile *et al.* (2015) analyzed SCFAs in a three-stage continuous fermentative colonic model through derivatization followed by GC analysis. Pérez-Burillo *et al.* (2019) analyzed microbiota from an *in vitro* digestion system by HPLC. One of our collaborators is optimizing their protocols for UPLC-MS/MS analysis to quantify SCFAs in gut microbiota cultured in our MiPro medium.

References:

- M Alander, I De Smet, L Nollet, *et al.* (1999), The effect of probiotic strains on the microbiota of the Simulator of the Human Intestinal Microbial Ecosystem (SHIME). *International Journal of Food Microbiology*, 46(1): 71-79.
- A Costabile, GE Walton, G Tzortzis, *et al.* (2015) Effects of orange juice formulation on prebiotic functionality using an *in vitro* colonic model system. *PLOS ONE* 10(3): e0121955.
- S Pérez-Burillo, T Mehta, A Esteban-Muñoz, *et al.* (2019) Effect of *in vitro* digestion-fermentation on green and roasted coffee bioactivity: The role of the gut microbiota. *Food Chemistry*, 279: 252-259.

3. The authors should determine the maximal run time for an experiment in this culturing approach.

Response: Our initial targeted application was for studying drug responses. Gut microbes can rapidly respond to the altered environmental factors. For example, Faith *et al.* (2011) and David *et al.* (2014) found rapid response of gut microbiota to diet within 1 day. In our own studies, we have also found that the gut microbiome responds quickly to different medium compositions (in 12 hr, Li *et al.*, 2018) and to most drugs (1 day, Li *et al.*, 2019). Therefore, for most drug screening a 24 hr culturing would be sufficient. Nonetheless, since our first submission, we have extended the culture to five days. To achieve longer culture, we modified the protocol by replacing one quarter of the cultured suspension (i.e. 250 μ l) with an identical volume of fresh medium every 12 hr. This replenish the nutrients and reduces the accumulation of microbial metabolites. Taxon-function-coupled analysis showed that the Pearson's correlation coefficient r of the taxon-specific functional profiles between the baseline and cultured microbiome were well maintained above 0.8 for 5days of growth (Figure 2F).

Species-level biomass distribution bar-chart also suggested that the microbiome was sustained throughout the 5-day culturing period (Figure 2G).

Figure R5 (Figure 2F-G in manuscript). Metaproteomics revealed taxonomic & functional composition stability over time. (F) Pearson's correlation coefficient r of taxon-specific functional profiles between the baseline inoculum and microbiome cultured in MiPro over a 5-day period. (G) Compositional bar chart showing species-level biomass distribution of a gut microbiome cultured in MiPro over a 5-day period.

References:

- JJ Faith, NP McNulty, FE Rey FE, JI Gordon (2011). Predicting a human gut microbiota's response to diet in gnotobiotic mice. *Science*, 333:101–104.
- L David, CF Maurice, RN Carmody, *et al.* (2014), Diet rapidly and reproducibly alters the human gut microbiome. *Nature*, 505(7484): 559–563.
- L Li, X Zhang, Z Ning, *et al.* (2018), Evaluating in vitro culture medium of gut microbiome with orthogonal experimental design and a metaproteomics approach. *Journal of Proteome Research*, 17(1):154-163.
- L Li, Z Ning, X Zhang, *et al.* (2019), RapidAIM: A culture- and metaproteomics-based Rapid Assay of Individual Microbiome responses to drugs, *bioRxiv*, doi: 10.1101/543256

Conclusion:

In-vitro culturing that restores the different complex features of a microbiota sample is really highly relevant. And the metaproteomics is a good starting point for proving that but there are several aspects missing before the paper can be published.

Response: Thank you for the positive opinion on our manuscript. We sincerely hope our responses and revisions could address your concerns.

Reviewer #2 (Remarks to the Author):

This work has been done to demonstrate the utility of a batch culture method for maintenance of ecosystem function for study in a high-throughput manner.

The authors claim that their method of culture of the gut microbiota in deep-well 96-well plates – the basis of the MiPro – is novel and that this method of culture maintains the composition of the microbiota over time, thus mimicking the gut microbiota function better than in bioreactor systems where flow conditions are applied to allow the culture to attain steady state. I agree that a limitation of more complex *in vitro* models is the change that the supported ecosystem has to undergo in order to cope with its new *in vitro* conditions (mostly these changes are related to substrate availability). However, I do not think that the MiPro system is actually a valid comparator. It is essentially a high throughput batch culture system, however the stability that it claims to have has only been measured over a maximum of 48 hrs. During this time, the cell mass within each well increased, but this is expected. The novelty seems to come from suggesting that the end result after just 48 hrs looks similar to the inoculum, but I think this would be true of any system where only 4 generations were followed, and I am struggling to see what is new here. The proteomics work done, while it is good and careful work, is likely not very representative of the new biomass, but of the original inoculum.

Response: We appreciate the opportunity to better clarify the rationale and novelty of our study. The intent of our approach is different from previous systems. Here we'd like to make the points clearer:

- 1) The focus of our paper is to provide conditions suitable for 96-well screening while maintaining the composition similar to the original inoculum and allowing individual microbiomes to be tested. This is particularly important considering the interest in screening for drugs that can modulate the microbiome and to test whether drugs not intended for the microbiome have off-target effects on the gut microbiome. A recent study in Nature (Maier *et al.*, 2018) has observed extensive impact of non-antibiotic drugs on cultured human gut bacterial strains by testing 1,200 approved drugs. However, response of isolated strains may be different from those in a functionally preserved gut microbiome because of the complex functionality of a microbial community, such as bacteria-bacteria interactions. Therefore, there's a pressing need of systematically studying drug effects using functionally maintained individual gut

microbiomes as an *in vitro* model in a system that can provide suitable throughput while representative of the inoculum.

- 2) A scalable 96-deepwell plate-based procedure would be suitable for such high-throughput drug screenings – the volume is enough for downstream analysis, and it's easily adaptable to automations. Fluidic models have the strength in long-term observation of xenobiotic effects, but would be difficult to adapt for high-throughput. We did not intend to imply that our 96-well-based culturing model is a valid comparator to those complicated fluidic systems as the purposes are very distinct. Our previous version of manuscript used an improper tone towards these models that may lead to misunderstanding, we have fixed it in the Introduction.
- 3) In short, the strength of our work lies in functionally maintained individual gut microbiota for systematic studying drug effects. We have clarified this point in the revised Introduction (page 2, lines 34-42), as follows.

“Moreover, a high percentage of marketed drugs, and compounds in development, may have off-target effects on the gut microbiome^{15,16}. Maier *et al.* observed that non-antibiotic drugs had extensive antibiotic-like impacts on cultured human gut bacterial strains¹⁷. However, the response of isolated strains may differ from that of a functionally preserved gut microbiome due to the complex functionality of a microbial community. Therefore, systematic studies of drug effects using functionally maintained individual gut microbiomes, as the *in vitro* model, is a pressing need. For long-term observations of xenobiotic effects, continuous flow systems^{12,18,19} and microfluidic models^{20,21} work well. However, these models can not be readily adapted for high-throughput approaches, partially due to their sizes and the time required for setting up and stabilizing these bioreactors.”

Using this model, we were able to reveal individualized microbiome functional responses to drugs (Li *et al.*, 2019a).

Our paper solves a major issue that exists in current *in vitro* culturing processes, which is the maintenance of gut microbiota composition and function following inoculation. Unlike other studies that selected a medium composition from existing media formulations, we previously did a careful study on the functional effect of different medium components on the gut microbiome using metaproteomics (Li *et al.*, 2018) and accordingly optimized our medium composition. In the current manuscript, we further optimized the bile salts composition.

There is evidence in the literature that current batch and flow culturing approaches lead to drastic changes in the microbiome within the first 48 hours (Batch: Li *et al.*, 2019b, Johnson *et al.*, 2015, Long *et al.*, 2015, Kim *et al.*, 2011; Flow: Auchtung *et al.*, 2015; McDonald *et al.*, 2013). We also showed in the current study that culturing in tubes for 24 hr, or altering the bile salt composition, caused marked shifts in the microbiota. However, after optimizing our culture medium and physical culture setup, we were able to stabilize the microbiome for 48 hrs.

Figure R6. Literature showing rapid change of gut microbiota profiles within 48 hrs. Batch culturing being used for in vitro microbiota studies showed shifted microbiome profiles, as can be clearly seen from composition bars (E Li *et al.*, 2019), heatmap (Long *et al.*, 2015), and PCR-DGGE bands (Kim *et al.*, 2011). Johnson *et al.* (2015) observed a significant shift along the major axis of variation after culturing for 10 hr, in a PCA based on the span of the whole HMP dataset. For continuous flow culturing, a previous study on the minibioreactor (Auchtung *et al.*, 2015) indicated that the approach can maintain

the microbiome from 3-21 days; however, the figure showed that within the first two days the microbiome diverged rapidly.

Furthermore, we have performed an additional experiment and have validated that the culturing period could be extended to up to five days (please refer to our response to Reviewer #1's last comment for details).

With regards to the comment about biomass, in our MiPro model, viable bacterial count was quintupled after 48 hours. This means that a high proportion of biomass used for the proteomics analysis was new (Figure R7). Therefore, we believe that our analysis is representative of the microbiome after 48 hours of culture, rather than just the inoculum.

Figure R7. Comparison of the abundance and composition of samples.

Reference:

- L Maier, M Pruteanu, M Kuhn, *et al.*, (2018), Extensive impact of non-antibiotic drugs on human gut bacteria, *Nature*, 555(7698):623-628.
- L Li, Z Ning, X Zhang, *et al.* (2019a), RapidAIM: A culture- and metaproteomics-based Rapid Assay of Individual Microbiome responses to drugs, *bioRxiv*, doi: 10.1101/543256
- L Li, X Zhang, Z Ning, *et al.* (2018), Evaluating in vitro culture medium of gut microbiome with orthogonal experimental design and a metaproteomics approach. *Journal of Proteome Research*, 17(1):154-163.
- E Li, H Yang, Y Zou *et al.*(2019b), *In-vitro* digestion by simulated gastrointestinal juices of *Lactobacillus rhamnosus* cultured with mulberry oligosaccharides and subsequent fermentation with human fecal inocula. *LWT*, 101:61-68.
- LP Johnson, GE Walton, A Psichas (2015). Prebiotics modulate the effects of antibiotics on gut microbial diversity and functioning *in vitro*. *Nutrients*, 7(6), 4480-4497.
- W Long, Z Xue, Q Zhang, *et al.* (2015). Differential responses of gut microbiota to the same prebiotic formula in oligotrophic and eutrophic batch fermentation systems. *Sci Rep* 5, 13469.

- BS Kim, JN Kim, and CE Cerniglia (2011), *In vitro* culture conditions for maintaining a complex population of human gastrointestinal tract microbiota. *J Biomed Biotechnol* 2011, 838040.
- JM Auchtung, CD Robinson and RA Britton (2015). Cultivation of stable, reproducible microbial communities from different fecal donors using minibioreactor arrays (MBRAs). *Microbiome*, 3:42
- JAK McDonald, K Schroeter, S Fuentes, *et al* (2013). Evaluation of microbial community reproducibility, stability and composition in a human distal gut chemostat model. *J Microbiol Methods* 95, 167.

If one also considers the work of Auchtung *et al.* (*Microbiome* 2015; 3:42), there is also really no novelty in the high throughput aspect of the current work, since this has been done before and, indeed, validated across different individuals.

Response: As we mentioned, the purpose of this article was to develop conditions suitable for high-throughput screening that would maintain to the original inoculum at both the functional and the compositional level. We have not only optimized the culture conditions but also added a perforated silicone-gel cover to allow positive pressure in wells. The perforated silicone-gel cover made a marked difference relative to tube culture (Figure 1B and Supplementary Figure S2). In contrast, the work by Auchtung *et al.* did not maintain the gut microbiota at its initial stability state (Figure R6 in our response letter), although the microbiota was stable from days 3-21, the first two columns representing days 1-2 (highlighted by blue arrows) are markedly different – most OTUs were almost reversed in color. Moreover, the profile of the day 0 inoculum wasn't shown. Our paper solves a major issue that exists in current *in vitro* culturing processes, which is the maintenance of the same gut microbiota composition from inoculation until the end of the experiment.

I am troubled by the use of OD as a measure of growth. This is a proxy for growth only, and cannot accurately be used for mixed species ecosystems where multiple species of highly variant cell size are to be considered (such as that of the gut microbiome). Since flow cytometry was carried out, why not use this to accurately count biomass?

Response: Although we agree that OD testing is only a proxy for growth, it is a well-accepted and robust tool to assess bacterial density (Broderick *et al.*, 2014). We used both approaches together to allow a comprehensive assessment; flow cytometry provides live and dead cell counts over time, and OD provides a verification that bacterial density is increasing, supporting the flow cytometry results.

Reference:

- NA Broderick, N Buchon, B Lemaitre (2014). Microbiota-Induced Changes in *Drosophila melanogaster* Host Gene Expression and Gut Morphology. *mBio*, 5(3):e01117-14.

I would like to see the results gained from tube culture of the 3 tested ecosystems presented over time and compared to the MiPro culture, as is presented for the MiPro culture in figure 2. Otherwise, only a global Firmicutes:Bacteroidetes ratio is shown in figure 1 – but this is not really enough to draw conclusions about the comparison between the 2 methods of culture.

Response: For a rigorous comparison of the two methods, we calculated the correlations between the taxon-specific functional profile of the inocula and that of the tested conditions (Figure R8 - new Figure 1B). In addition, we did taxonomic and functional analysis (Figure R9 - new Supplementary Figure S2) as presented for the MiPro culture in Figure 2. After 24 hours of culturing, the pre-test results suggest significantly higher performance of the CDCA+CA & 96-well culturing condition. We don't think that it's necessary to do another validation on the three ecosystems overtime, because the difference was clear by 24 hours and because the 24-hour validation was simply a pre-test for the follow-up *in-vitro* and *in-vivo* validation experiments.

Figure R8 (Figure 1B in manuscript). Pearson's correlation coefficient r of taxon-specific functional profiles between the inocula (0 hr baseline sample) and 96-deep well cultured microbiome with the presence of primary bile salts (CDCA + CA) or commercialized bile salts mixture (DCA + CA), as well as tube-cultured microbiome with the presence of primary bile salts. Different letters indicate significant differences at the $p = 0.05$ level, Tukey-b test; box spans interquartile range (25th to 75th percentile), and line within box denotes median. Whiskers represent min to max values).

Figure R9 (Figure S2 in manuscript). Pre-test for selecting optimal bile salts composition and culture conditions of the gut microbiome. (A) Pre-experiment showing the relative abundances of Firmicutes and Bacteroidetes in the inocula (0 hr baseline sample), 96-deep well cultured microbiome with the presence of primary bile salts (CDCA + CA) or commercialized bile salts mixture (DCA + CA), as well as tube-cultured microbiome with the presence of primary bile salts. (B) Compositional bar chart showing species-level biomass distribution after 24 hr culturing under the three treatments. (C) PCA scores plot with hierarchical clustering based on abundances of COG functional categories of the microbiome sample cultured under the three treatments. Samples were analyzed on an Orbitrap XL following a 6 hr gradient. 24,005 peptide sequences corresponding to 6,301 protein groups were identified with an average MS/MS identification rate of $17.6\% \pm 2.6\%$ (mean \pm SD).

The strength of the manuscript lies in the approach used to compare samples through metaproteomics (although not in itself a novel approach) and the testing of different bile salt mixtures on the microbial ecosystems, both of which are useful additions to the field. However, I would like to see more evidence that this MiPro model is truly unique in its ability to support ecosystems while maintaining inocula specific characteristics – to do this would require following the system for at least 72 hrs and possible even longer. Otherwise, all that this system

represents is a simple, and not really all that novel, batch culture system for the gut microbiome that only really has any utility for short-term experiments and drug interaction studies.

Response: To verify that the MiPro model is able to support ecosystems while maintaining inocula-specific characteristics, we have performed an additional experiment confirming that the microbiome remains stable for at least five days of culture (please refer to our response to Reviewer #1's last comment for details).

We think that continuous flow system and high-throughput batch culturing are equally important, and meet different research and application demands. Although continuous flow systems allow comprehensive study of xenobiotics' effects on the gut microbiota, it would be impractical to test thousands of drugs in a fluidic system. Gut microbes can rapidly respond to the altered environmental factors, so short-term studies are extremely useful for drug interaction screening.

Reviewer #3 (Remarks to the Author):

General comments:

This is a well-designed study that addresses the important issue of developing in vitro systems for culturing and testing complex bacterial populations such as the gut microbiome. Correlating the in vitro results from the authors' MiPro method with in vivo results is well done.

The Results section and the Methods section are, in general, well written for both clarity of thought and language. The Introduction and Discussion sections (and to a lesser degree the Abstract) appear to have been written by a different author and may suffer from the challenge of writing in English as a second language. Several specific examples are pointed out below, but these sections should be read and edited by the other authors to obtain a uniformly high level of presentation throughout the manuscript.

Response: We would like to thank Reviewer #3 for the thoughtful comments. We have improved the Introduction, Discussion and Abstract, and have asked native English speaking researchers to edit the manuscript. To distinguish from revisions of contents which were marked in blue, we used track changes for language edits.

Specific points:

1. The authors describe the MiPro system as suitable for high-throughput investigations. While the method scales well for sample replication and multiple testing conditions, the length of analysis time per sample (90-120 minute gradients for peptide separation) suggests that a single 96-well plate would require more than 6 days of LC-MS/MS instrument time alone. Calling this “high-throughput” is arguably little strong.

Response: The reviewer’s point is well taken and we have modified the language in the manuscript to reduce the emphasis on high-throughput screening. It’s true that the current metaproteomics process that we use is a speed limiting step if we use it to analyze microbiome’s drug responses; it may also be true when using other -omics technologies. We are working to add multiplexing techniques, such as iTRAQ, to increase the throughput of the metaproteomic analysis.

2. On page 4 the authors state that they “observed a 4.4-fold increase of viable bacterial count in MiPro medium after 24 hr of culturing, whereas a 3.0-fold maximum increase was detected in the BCM medium at 9 hr post-culturing.” It is not clear how these fold increases were derived from the data presented in Figure 1. A better description – either in the Results or Methods section – is necessary.

Response: We have clarified how the data were derived in the Results (page 4, lines 105-110). In addition, we noticed that there were currently two ways of describing “fold-increase” in literatures. Often in omics studies, for quantities A and B, the fold change of B with respect to A is described as B/A . An alternative definition which is mathematically more rigorous is given as $(B - A)/A$. In this paper we used the latter. To also clarify this, we added a description in the Methods section (page 17, lines 422-427), marked in blue, as follows:

In Results (page 4, lines 105-110): “Furthermore, using flow cytometry in combination with viability staining, we found that our MiPro medium achieved high ratio of viable bacterial cells throughout the culture period (95.8% at 24 hr compared with 71.2% at 0 hr, Figure 1D). Comparison of the flow cytometry readouts between the inoculum and the cultured microbiome showed a 4.4-fold increase of viable bacterial count in MiPro medium after 24 hr of culturing”

In Methods, Page 17, lines 422-427: “For each sample, data were recorded for 2 minutes on the low flow rate setting. Bacterial cells were gated according to size and granularity on the FSC/SSC scatter plot. Live/dead cells were gated according to the gates set by the

stained gut microbiome, the heat-treated and stained gut microbiome, and the unstained gut microbiome (Supplementary Figure S3). Data were analyzed using Kaluza Analysis Software version 1.5. Live-gated bacterial cell counts were compared for its fold change in the cultured microbiome with respect to the inoculum, i.e. (cultured – inoculum)/inoculum.”

3. In the Results section on page 3, the authors suggest that the superiority of the deep well plate culturing method over culture tubes may be due to reduced gas-exchange with the outer environment and preservation of partial pressure of gasses and volatile metabolites. While it isn't immediately clear why a perforated membrane would allow less gas transfer than a loosened cap, this explanation is at least plausible. However, on page 12 in the Discussion section, the authors state the above explanation as establish fact. If you have done any gas analysis of the headspace to prove this, it isn't presented or described. There are other possible explanations for the difference (i.e. differences in surface area to volume in the cultures, agitation mechanics in differently-shaped culture vessels, or others). The statement in the Discussion should be written as a possibility – not fact.

Response: We have modified the language in the Discussion to make it more clear that this is a speculation (page 13, lines 309-317):

“One possible explanation could be that the silicon-gel cover may have increased the partial pressure of gases produced by the gut microbiome, such as H₂, CH₄, H₂S and NO_x, etc.⁵⁴ and volatile organic compounds, such as SCFAs⁵⁵. This could preserve the levels of dissolved gas in the culture medium. Some dissolved metabolites are important factors for bacteria cross-feeding⁵⁵ and the control of pathogens⁵⁶, and thus, are presumably essential for the maintenance of an *in vitro* microbiome. Other factors may include differences in the surface area to volume ratio, agitation mechanics from cultures in differently-shaped vessels. Our model was developed for rapidly screening drugs against different individual's gut microbiomes⁵⁷.”

4. Page 6: On one line the samples from the MiPro are described as “OM-cultured samples.” This was the only use of “OM-cultured” that I could find in the manuscript. Legacy from an earlier version? Either define the term or change it to be consistent with the rest of the manuscript.

Response: We have corrected “OM” to “MiPro”.

5. In the legend for Figure 1, the description for panels C and D are mislabeled as B and C, respectively.

Response: We have corrected the labels.

6. In Figure 2 panel E, the COGs 0149, 3842, and 0052 in the 0 hr diagram are not paired with pathways in the 24 hr diagram. Shouldn't these be in gray text as well?

Response: We have changed the colors of these COGs into gray text.

Figure R10: revised Figure 2E

7. The first two paragraphs of the Discussion section have several sentences duplicated verbatim. Choose one location or the other.

Response: We have deleted the duplicated sentences.

8. The phrase “time-and-cost saving solutions” (Abstract [page 1] and Introduction [page 2]) is atypical and grammatically questionable.

Response: We have corrected the phrase in the Abstract and Introduction.

9. There are multiple misspellings or incorrect word choices scattered throughout the manuscript (e.g. equal should be equate [page 2], alternations should be alterations [page 11], health should be healthy and recapitulate should be recapitulate or replicate [page 13], etc.). Careful reading and editing is needed throughout.

Response: We have corrected all the words mentioned by Reviewer #3, and have asked native English speaking authors to have the whole manuscript carefully read and edited.

The authors present interesting and useful results, particularly with respect to the bile acid composition of the growth medium, the effect of different culture vessels, and the ability of the

assay to mirror *in vivo* results under the conditions they tested. There are some potential limitations to the MiPro approach, however. For example, the use of a dilute initial bacterial concentration and dependence of the method on bacterial growth in a 24 hour period raises the possibility of masking the contributions of bacterial species with slower growth rates. Also the 24 hour window may be too short for some types of drug or xenobiotic exposure studies. Using a metaproteomic approach to the analysis to generate both biomass and function data is efficient, but does it introduce biases in bacterial identification that could create problems? The Discussion and Conclusion sections tout the benefits of the method, but some additional discussion of the potential limitations should be included as well.

Response: We have added a paragraph in the Discussions discussing our limitations of this study, including those mentioned here (page 15, lines 365-371), as follows:

“Although the composition and functions were maintained for each individual microbiome, our study was not designed to detect changes in very low abundance bacterial species. In addition, factors such as different protein extraction protocols and bioinformatic database may introduce sources of bias in metaproteomic analysis⁶⁴. Further in-depth investigations on *in vitro* culturing and drug responses could combine metaproteomics with other omics technologies such as metagenomics and metabonomics to gain deeper insights of the microbiome structure and function.”

Although we and others have shown that the gut microbiome responds quickly to most environmental changes, as discussed above, the reviewer is correct that some xenobiotics may act more slowly. We have therefore performed an additional experiment to validate extension of the culturing period up to five days (please refer to our response to Reviewer #1’s last comment for details).

REVIEWERS' COMMENTS:

Reviewer #2 (Remarks to the Author):

I appreciate the efforts made by the authors to clarify their work and the justification for the MiPro system. These efforts have made the work much clearer and more understandable. I think that the decision to extend the run time to 5 days is commendable, and makes the work much more valuable.

I still have some concerns:

Figures R4 and R5 need to be normalised to cell count before they can be reliably used to make comparisons across samples. Since cell counts were made with flow cytometry, this should be fairly straightforward and will give a more accurate picture of what is going on, and may actually help to better support the premise of the work.

The authors state in their rebuttal that "We used both approaches together to allow a comprehensive assessment; flow cytometry provides live and dead cell counts over time, and OD provides a verification that bacterial density is increasing, supporting the flow cytometry results."

My point here was to ask why OD was done at all, when flow cytometry - a much more sensitive and quantitative technique, much better suited to mixtures of different cell sizes, was carried out. Surely flow cytometry data is considered more accurate, thus if this was done, why report on OD? There is an opportunity missed here to talk about the number of live and dead cells of different sizes across the spectrum of what is present in the gut microbiota samples, and how these increase or decrease over time. The Broderick et al. reference given in the rebuttal does not seem to be particularly relevant to this discussion (using OD to measure food content in guts of drosophila, and also considers only a very simple gut microbiota of similarly-sized cells) and I am puzzled by the authors' use of it.

Reviewer #4 (Remarks to the Author):

The manuscript describes a novel in vitro method to cultivate gut microbiome samples and maintain both functional and taxonomical profiles similar to in vivo gut microbiomes. The authors developed a 96-deep well plate culturing model (MiPro) and validated its ability to maintain the features of in vivo gut microbiomes by metaproteomics.

Several major concerns were pinpointed during the first revision:

- 1) The short running time of the experiments (48H) and questions surrounding the ability of the system to maintain the features of the samples for longer time.
- 2) Issues inherent to the selected approach (metaproteomics) as a basis to validate their model such as: a) Not really a high-throughput methods (6 days of instrument time for a 96-well plate); b) Metaproteomics is ideal for functional profiles, it is not the most suitable approach for taxonomic analyses and 16S rRNA analyses are the gold standard.
- 3) The novelty of the approach, notably in comparison with bioreactors (continuous flow systems) and fluidic systems (microfluidic models).

During the revision process, the authors have addressed these major concerns:

- 1) They have extended the culture time to 5 days and verified the sustainability of the system. While the Pearson's correlation coefficient of the taxon-specific functional profiles between the baseline and cultured microbiome decreased over time, it was still over 0.8 after 5 days. The authors also clarified their claims regarding the throughput of their approach and mentioned its current optimization.

Additional comments: Using longer culturing time, the authors identified more species (149) than with their first analysis based on 48H, where they identified 121 species (using similar threshold of at least 3 peptides). A longer cultivation time might allow more species to grow (slow growing species) and thereby allows a more comprehensive capture of the microbial diversity.

2) The authors argued that MS-based metaproteomics can accurately quantify taxon-specific biomass contributions and taxon-specific functional profiles. Nevertheless, they agreed that 16S rRNA gene sequencing is a gold standard and performed the analysis on backup samples for the different samples (0, 24 and 48 hours) and showed that 16S results supported their metaproteomic data.

Additional comments: Metaproteomics is indeed able to identify and quantify taxa but is strongly dependent on the database and the approach used. Running a metagenomic or 16S rRNA gene analysis was necessary. They could have even extended the analysis to the 2, 3, 4 and 5 days samples as it would have added an extra layer of information to support their results.

3) The concerns about novelty are more difficult to evaluate. However, in their revision, the authors have clarified their points and discussed the strength of their work. Evidence in the literature (Fig R6) showed that current batch and flow culturing approaches lead to changes in microbiota within the first 48 hours while the authors' system sustains the functional and taxonomic profile during the same time. As gut microbiome can respond to changes within a day, a 24H culture using the Mipro model would be applicable for drug screening, while other systems would be better for long-term observations and xenobiotic effects.

Supplementary comments:

Stability of the functional activities was assessed using proteins annotated with COG categories. It should be noted that aggregation of functions/proteins into broad functional orthology-based categories could contribute to an impression of stability. E.g. the proteins/functions are different, but they belong to the same category so it looks like nothing changed. "Finer" differences and so more reliable comparison could have been achieved using the individual protein annotation instead of using functional categories.

Line 45 a meta-omicS tool

Line 55: predicted from metagenomicS analyses

Line 56: Therefore, metaproteomics is a suitable tool

Line 84: abundances of clusters

Line 227: PLS-DA is first used at line 227 but is defined at line 252.

Figure 3D is not mentioned in the text.

Line 230: VIP score is not defined in the text. Only in the legend of the figure 3.

Line 243: at days 14 and 28. (n = 5).

Line 251 in vivo untreated vs. in vivo day 14 or day 28 (replace by versus. At least you used versus fully written in other places).

Signed: Paul Wilmes

Point-to-point response to reviewers' comments

Reviewer #2 (Remarks to the Author):

I appreciate the efforts made by the authors to clarify their work and the justification for the MiPro system. These efforts have made the work much clearer and more understandable. I think that the decision to extend the run time to 5 days is commendable, and makes the work much more valuable.

I still have some concerns:

1. Figures R4 and R5 need to be normalised to cell count before they can be reliably used to make comparisons across samples. Since cell counts were made with flow cytometry, this should be fairly straightforward and will give a more accurate picture of what is going on, and may actually help to better support the premise of the work.

Response: We would like to thank Reviewer #2 for the further suggestions to improve our manuscript. We agree with Reviewer #2's comment that we need to also show cell counts in addition to the relative composition. We were missing a figure to show the increased number of bacteria, therefore we first added a figure showing cell counts of live and dead bacteria beside our previous flow cytometry figure (Figure R11).

Figure R11 (Figure 1D and 1E in manuscript). (D) Temporal microbiome viability changes as shown with flow cytometry. The gating strategy is shown in Supplementary Figure S3. (E) Bacterial cell count in each flow cytometry recording. Data were recorded for 2 minutes on the low flow rate setting.

In addition, according to the reviewer's suggestion, we generated stacked column bar chart of species-level biomass composition normalized to cell counts, as shown in Figure R12 below. Considering that due to the lower cell count at the 0 hr, when showing the figure as below, it's difficult to compare the composition between 0 hrs and the other time points. So we would like to still keep the figures of relative abundance in the main contents, and add the figure normalized to cell count to the supplementary figure (Supplementary Figure 6).

Figure R12 (Supplementary Figure 6 in manuscript). Compositional bar chart of species-level biomass normalized to cell counts according to the flow-cytometry results.

We hope that these changes would together give a clearer picture that the microbiome composition was maintained along with a marked increase of bacterial number.

- The authors state in their rebuttal that "We used both approaches together to allow a comprehensive assessment; flow cytometry provides live and dead cell counts over time, and OD provides a verification that bacterial density is increasing, supporting the flow cytometry results."

My point here was to ask why OD was done at all, when flow cytometry - a much more sensitive and quantitative technique, much better suited to mixtures of different cell sizes, was carried out. Surely flow cytometry data is considered more accurate, thus if this was done, why report on OD? There is an opportunity missed here to talk about the number of live and dead cells of different sizes across the spectrum of what is present in the gut microbiota samples, and how these increase or decrease over time. The Broderick et al.

reference given in the rebuttal does not seem to be particularly relevant to this discussion (using OD to measure food content in guts of drosophila, and also considers only a very simple gut microbiota of similarly-sized cells) and I am puzzled by the authors' use of it.

Response: We would like to thank the reviewer for the thoughtful comment. Accordingly, we have added a discussion on why we used both methods in the text, marked in blue, as below (page 4, lines 102-104). We are sorry about citing the wrong reference in the rebuttal.

“Since different microbial members can differ by several orders of magnitude in biomass²⁸, it is necessary to examine both biomass and bacterial counts. We used optical density at 595 nm (OD₅₉₅) to determine bacterial biomass density, and used flow cytometry to determine bacterial cell counts.”

Reviewer #4 (Remarks to the Author):

The manuscript describes a novel in vitro method to cultivate gut microbiome samples and maintain both functional and taxonomical profiles similar to in vivo gut microbiomes. The authors developed a 96-deep well plate culturing model (MiPro) and validated its ability to maintain the features of in vivo gut microbiomes by metaproteomics.

Several major concerns were pinpointed during the first revision:

- 1) The short running time of the experiments (48H) and questions surrounding the ability of the system to maintain the features of the samples for longer time.
- 2) Issues inherent to the selected approach (metaproteomics) as a basis to validate their model such as: a) Not really a high-throughput methods (6 days of instrument time for a 96-well plate); b) Metaproteomics is ideal for functional profiles, it is not the most suitable approach for taxonomic analyses and 16S rRNA analyses are the gold standard.
- 3) The novelty of the approach, notably in comparison with bioreactors (continuous flow systems) and fluidic systems (microfluidic models).

During the revision process, the authors have addressed these major concerns:

- 1) They have extended the culture time to 5 days and verified the sustainability of the system. While the Pearson's correlation coefficient of the taxon-specific functional profiles between the baseline and cultured microbiome decreased over time, it was still over 0.8 after 5 days. The authors also clarified their claims regarding the throughput of their approach and mentioned its current optimization.

Additional comments: Using longer culturing time, the authors identified more species (149) than with their first analysis based on 48H, where they identified 121 species (using similar threshold of at least 3 peptides). A longer cultivation time might allow more species to grow (slow growing species) and thereby allows a more comprehensive capture of the microbial diversity.

Response: We would like to thank Reviewer #4 for the time and efforts in reviewing our manuscript. We appreciate the reviewer's careful thoughts about species diversity. We examined the five-day culture data and did not find significant changes (*t* test) in the alpha diversity over time (Figure R13). The different number of species could be because of the use of different stool samples in the two experiments.

Figure R13. Alpha diversity (Shannon-Weiner index) of samples during the five-day culturing.

2) The authors argued that MS-based metaproteomics can accurately quantify taxon-specific biomass contributions and taxon-specific functional profiles. Nevertheless, they agreed that 16S rRNA gene sequencing is a gold standard and performed the analysis on backup samples for the different samples (0, 24 and 48 hours) and showed that 16S results supported their metaproteomic data.

Additional comments: Metaproteomics is indeed able to identify and quantify taxa but is strongly dependent on the database and the approach used. Running a metagenomic or 16S rRNA gene analysis was necessary. They could have even extended the analysis to the 2, 3, 4 and 5 days samples as it would have added an extra layer of information to support their results.

Response: We again agree that metagenomic or 16S rRNA gene analysis are “gold standards” for microbiome membership composition, and we would definitely do these analyses in our future works when necessary. In our current study, we mainly focused on taxon-specific functional activities of the gut microbiome. Since functions predicted from 16S rDNA or

metagenomics analyses are not necessarily expressed, we chose metaproteomics as our major tool, which provides quantified protein abundances that estimate the functional activities of microbiome members.

3) The concerns about novelty are more difficult to evaluate. However, in their revision, the authors have clarified their points and discussed the strength of their work. Evidence in the literature (Fig R6) showed that current batch and flow culturing approaches lead to changes in microbiota within the first 48 hours while the authors' system sustains the functional and taxonomic profile during the same time. As gut microbiome can respond to changes within a day, a 24H culture using the Mipro model would be applicable for drug screening, while other systems would be better for long-term observations and xenobiotic effects.

Response: We would like to thank Reviewer #4 for the thoughtful comment.

Supplementary comments:

Stability of the functional activities was assessed using proteins annotated with COG categories. It should be noted that aggregation of functions/proteins into broad functional orthology-based categories could contribute to an impression of stability. E.g. the proteins/functions are different, but they belong to the same category so it looks like nothing changed. "Finer" differences and so more reliable comparison could have been achieved using the individual protein annotation instead of using functional categories.

Response: In the functional stability assessment results, we did both general and detailed functional comparisons. We firstly annotated the proteins to the COG categories to show a general, broad functional orthology-based stability. Next, through individual protein annotation, we did a taxon-function coupled analysis ---- we not only annotated the proteins to detailed functions (COGs), but also made each functional comparison taxon-specific. We generated a matrix of 1,066 unique COGs of proteins corresponding to 419 taxa for each sample (Supplementary Figure 1), and did the comparisons of taxon-specific functional activities between samples on this basis.

Line 45 a meta-omicS tool

Response: We have corrected this typo.

Line 55: predicted from metagenomicS analyses

Response: We have corrected this typo.

Line 56: Therefore, metaproteomics is a suitable tool

Response: We have corrected this typo.

Line 84: abundances of clusters

Response: We have corrected this typo.

Line 227: PLS-DA is first used at line 227 but is defined at line 252.

Response: We have moved the definition of PLS-DA to page 9, lines 229-230.

Figure 3D is not mentioned in the text.

Response: We have added Figure 3D in the text.

Line 230: VIP score is not defined in the text. Only in the legend of the figure 3.

Response: We have added a definition of VIP in page 10, line 233.

Line 243: at days 14 and 28. (n = 5).

Response: We have corrected this typo.

Line 251 in vivo untreated vs. in vivo day 14 or day 28 (replace by versus. At least you used versus fully written in other places).

Response: We have replaced vs. by versus.

Signed: Paul Wilmes